# PROTAC-induced protein structural dynamics in targeted protein degradation

**Kingsley Y Wu[1†], Ta I Hung[1,2†], Chia-en A Chang[1]\***

[1]Department of Chemistry, University of California, Riverside, Riverside, United States; [2]Department of Bioengineering, University of California, Riverside, United States

## eLife Assessment

This study provides **important** computational insights into the dynamics of PROTAC-induced degradation complexes, offering a **convincing** demonstration that differences in degradation efficacy can be linked to linker properties. The analyses address reproducibility considerations comprehensively, reinforcing the study's conclusions. Overall, these findings are significant for advancing cancer treatments and will be of broad interest to both biochemists and biophysicists.

**\*For correspondence:**
chiaenc@ucr.edu

[†]These authors contributed equally to this work

**Competing interest:** The authors declare that no competing interests exist.

**Abstract** PROteolysis TArgeting Chimeras (PROTACs) are small molecules that induce target protein degradation via the ubiquitin-proteasome system. PROTACs recruit the target protein and E3 ligase; a critical first step is forming a ternary complex. However, while the formation of a ternary complex is crucial, it may not always guarantee successful protein degradation. The dynamics of the PROTAC-induced degradation complex play a key role in ubiquitination and subsequent degradation. In this study, we computationally modelled protein complex structures and dynamics associated with a series of PROTACs featuring different linkers to investigate why these PROTACs, all of which formed ternary complexes with Cereblon (CRBN) E3 ligase and the target protein bromodomain-containing protein 4 (BRD4[BD1]), exhibited varying degrees of degradation potency. We constructed the degradation machinery complexes with Culling-Ring Ligase 4A (CRL4A) E3 ligase scaffolds. Through atomistic molecular dynamics simulations, we illustrated how PROTAC-dependent protein dynamics facilitating the arrangement of surface lysine residues of BRD4[BD1] into the catalytic pocket of E2/ubiquitin cascade for ubiquitination. Despite featuring identical warheads in this PROTAC series, the linkers were found to affect the residue-interaction networks, and thus governing the essential motions of the entire degradation machine for ubiquitination. These findings offer a structural dynamic perspective on ligand-induced protein degradation, providing insights to guide future PROTAC design endeavors.

## Introduction

With the rapid progressive efforts from modern drug discovery and development, numerous promising paradigms for disease treatment are emerging from the wealth of medicinal chemistry and biology. A comprehensive understanding of the molecular mechanisms underlying the function of targeted biological systems expedites the drug development process. Among these innovative approaches, PROteolysis TArgeting Chimeras (PROTACs), also known as hetero-bifunctional degraders, stand out as a revolutionary paradigm for selectively degrading a diverse array of disease-associated targets (*Dale et al., 2021*; *Békés et al., 2022*; *Ramachandran and Ciulli, 2021*). The formation of a ternary complex involving a PROTAC binding to an E3 ligase and its neo-substrate, commonly referred to as the protein of interest (POI), triggers the cell's native protein degradation machinery (e.g.

ubiquitination), which marks the target protein for proteasomal degradation. Previous studies have highlighted the importance of forming a high binding-affinity, stable and long-lived E3-PROTAC-POI complex for achieving potent degradation (*Bondeson et al., 2018*). However, it is noteworthy that the formation of a stable ternary complex does not always correlate with the degradation potency of the POI (*Bondeson et al., 2018*; *Huang et al., 2018*; *Nowak et al., 2018*).

The final degradation of a target protein, involving the three-step ubiquitination process (activation-E1, conjugation-E2, and ligation-E3) is facilitated by the binding a ternary complex induced by a PROTAC. Recent studies used E3 ligases such as MDM2, IAP, RNF4, and βTRCP to degrade various biological targets (*Zhao et al., 2019*; *Tinworth et al., 2019*; *Ward et al., 2019*; *Ottis et al., 2017*). While stable ternary complexes are often associated with high POI degradation efficiency, discrepancy between stable ternary complexes and degradation have also been observed. For instance, certain von Hippel Lindau (VHL)-based PROTACs exhibit high binding affinity to POIs but fail to induce degradation despite forming stable ternary complexes. Conversely, proteins like p38α, which exhibit low binding affinity with VHL-based PROTACs, undergo rapid degradation within a short timeframe (*Bondeson et al., 2018*; *Smith et al., 2019*; *Lai et al., 2016*). Therefore, it is evident that the formation of a stable ternary complex does not guarantee high degradation efficiency. Recent studies have suggested that the accessibility of lysine (Lys) residues of the POI after the formation of stable ternary complexes is critical for degradation and requires further research (*Bondeson et al., 2018*; *Han, 2020*; *Bai et al., 2022*).

The formation of a stable E3-PROTAC-POI complex represents the initial crucial step for POI degradation. Several modeling-based approaches have been developed to optimize and rationalize the PROTAC-mediated ternary complex. For example, several studies have employed protein–protein docking, linker conformational searching (*Drummond and Williams, 2019*; *Drummond et al., 2020*; *Zaidman et al., 2020*; *Bai et al., 2021*), and machine learning approaches (*Rao et al., 2023*; *Zheng et al., 2022*) to generate ensembles of E3-PROTAC-POI complexes in silico. Recent advancement in docking algorithms aimed to provide higher accuracy and energetically favorable ternary complexes (*Ignatov et al., 2023*; *Pereira et al., 2023*). Notably, recent studies have suggested that, in addition to forming a ternary complex, the orientation of E3-PROTAC-POI allowing accessibility of the surface Lys residues of the POI are important for ubiquitination. Because the subsequent ubiquitination process occurs in a ligase complex comprising multiple proteins, recent research efforts have begun to investigate the assembly of the E3-PROTAC-POI complex with the entire ligase complex and key enzymes E2/ubiquitin (E2/Ub) for ubiquitination. For example, two studies applied docking and structure-based modeling approaches to model CDKs-PROTACs and BCL-xl-PROTACs in a Cereblon (CRBN)-based Culling-Ring Ligase 4 A (CRL4A) complex and a VHL-based CRL complex, respectively (*Bai et al., 2022*; *Lv et al., 2021*). These studies integrated experimentally determined protein structures to construct a degradation complex using protein structure alignment approaches, and the distance measurements between Lys residues and Ub to show potential ubiquitination. Moreover, enhanced sampling molecular dynamics (MD) techniques have also been employed to investigate the spatial organization of POI within the context of a ligase complex for ubiquitination (*Dixon et al., 2022*). Nevertheless, challenges remain in accurately modelling the structural dynamics of the degradation machinery complex, and gaining a deeper understanding of the steps involved in POI ubiquitination would aid PROTAC design.

Several existing studies have examined a set of PROTACs targeting the same E3 ligase and POI for structure–activity relationships (*Liu et al., 2022*; *Bricelj et al., 2021*; *Lee et al., 2022*). Many of these studies have maintained consistent binders for the E3 and POI while varying the linker regions. For example, studies of several PROTACs (i.e. small-molecule degrader of bromodomain and extra-terminal domain [dBET] family) designed for degrading bromodomain-containing protein 4 (BRD4[BD1]) utilized the same BRD inhibitor (i.e. thieno-triazolo-1,4-diazepine [JO1]) and E3 ligase inhibitor, pomalidomide (or lenalidomide and thalidomide) while varying the linkers (*Nowak et al., 2018*; *Zhou et al., 2018*; *Lu et al., 2015*; *Filippakopoulos et al., 2010*). Although the crystal structures of these CRBN-dBETs-BRD4[BD1] ternary complexes have highly similar CRBN-BRD4[BD1] contacts, significant differences in degradation capabilities (DC$_{50/5h}$) have been observed (*Nowak et al., 2018*; *Zhou et al., 2018*; *Winter et al., 2015*; *Qin et al., 2018*). These findings underscored the need to further understanding of how the linkers and the formation of stable ternary complex may influence degradation capability. Although the linker region may result in minor differences in ligand solubility and/or membrane

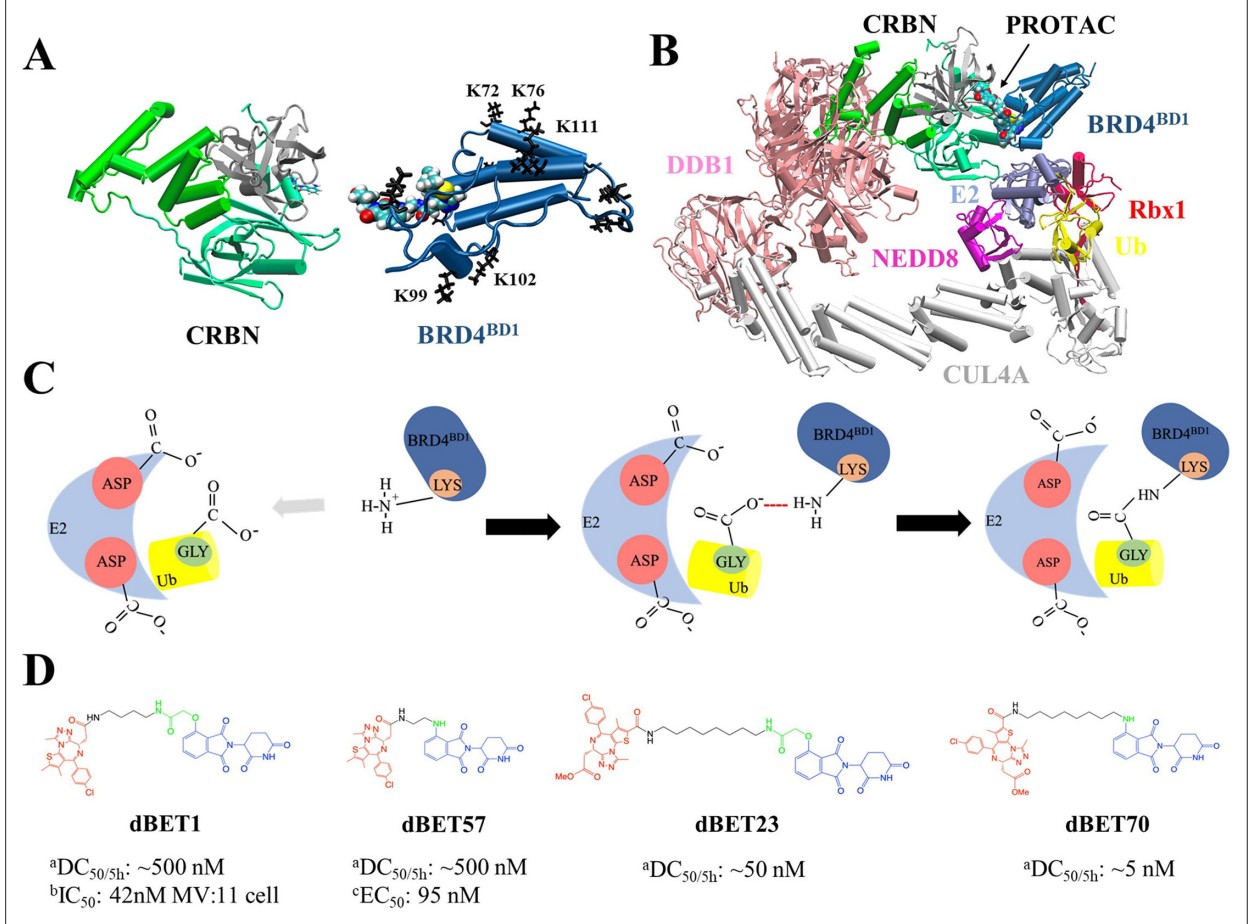

**Figure 1.** Structures of PROteolysis TArgeting Chimera (PROTAC)-mediated degradation machinery complex. (**A**) Cereblon (CRBN) E3 ligase, consists of an N-terminus domain (mint), helical binding domain (green), and C-terminus domain (gray). Notably, CRBN E3 ligase is also simply termed CRBN in the main text. Bromodomain-containing protein 4 (BRD4$^{BD1}$; targeted protein) consists of several important Lys residues (K72/76/99/102/111 black sticks) for successful degradation. (**B**) Degradation machinery complex is constructed with DDB1, CUL4A, NEDD8, Rbx1, E2 enzyme, ubiquitin (Ub), and CRBN-PROTAC-BRD4$^{BD1}$. Note, the complex without PROTAC recruited BRD4$^{BD1}$ is termed CRLA4A E3 ligase. (**C**) An illustration of the theoretical model of ubiquitination reaction. The catalytic site with multiple Asp residues creates a negatively charged environment that attracts Lys residues to enter the catalytic site. Lys and Asp residues then react with the C-terminus Gly of Ub for future degradation. (**D**) Chemical structure of each dBET PROTAC. [a]Degradation profile DC$_{50/5h}$ for four PROTACs was obtained from EGFP/mCherry reporter assay published in *Nowak et al., 2018*. [b]Data published in *Qin et al., 2018*. [c]Data published in *Nowak et al., 2018*.

The online version of this article includes the following figure supplement(s) for figure 1:

**Figure supplement 1.** Conformational ensembles of ternary complexes (CRBN-degrader of bromodomain and extra-terminal domain (dBETx)-BRD4$^{BD1}$) from protein–protein docking.

**Figure supplement 2.** Superpositions of predicted CRBN-dBETx-BRD4$^{BD1}$ ternary complexes (red for dBET23 and cyan for dBET70) with reported crystal structures (Black).

permeability, the proximity and accessibility of surface Lys residue(s) of the POI to the E2/Ub cascade in the ubiquitination site can be due to different linkers (*Smith et al., 2019*; *Zhang et al., 2022*). Thus, investigating the structures of the degradation complexes with PROTAC-induced ubiquitination while considering protein structural dynamics provides valuable insights into PROTAC degradability.

In this study, we employed protein–protein docking, structural alignment, atomistic MD simulations, and post-analysis to model a series of CRBN-dBET-BRD4$^{BD1}$ ternary complexes and the entire degradation machinery complex consisting of BRD4$^{BD1}$ and a CRL4A E3 ligase scaffold (*Figure 1*). These degraders, with different linker properties, were all capable of forming stable ternary complexes, but exhibited different degradation capabilities (*Nowak et al., 2018*; *Zhou et al., 2018*; *Winter et al., 2015*; *Qin et al., 2018*). The best degrader, dBET70, had a DC$_{50/5h}$ of about 5 nM, followed by dBET23 (DC$_{50/5h}$~50 nM) (*Figure 1D* and *Figure 1—figure supplement 1*). Although utilizing the

exact same warheads, other degraders such as dBET1 and dBET57 had a DC$_{50/5h}$ of about 500 nM. Our studies identified structural features of the dBETs that contribute to large-scale protein motions and explained the connection between the cellular activities and degradability reported for the dBETs with atomics details. Because finding energetically stable ternary conformations is a critical first step for protein degradation (*Nowak et al., 2018*; *Zaidman et al., 2020*; *Bai et al., 2021*), in addition to protein–protein docking, we also performed MD simulations to thoroughly cover the conformational space and performed energy calculations to ensure that our modeled ternary complexes were thermodynamically stable.

Assembling these stable CRBN-dBETx-BRD4$^{BD1}$ complexes into multiple modeled CRL4A E3 ligase-based scaffold conformations, we first observed that no surface Lys residue(s) of BRD4$^{BD1}$ was ready for the next ubiquitination step in the modeled degradation machinery complexes. Nevertheless, our unbiased MD simulations illustrated protein structural dynamics of the entire complex and local side-chain arrangements to bring Lys residue(s) to the catalytic pocket of E2/Ub for reactions. Post-analysis revealed the essential motions of the degradation complex and interactions crucial for ubiquitination. Our results relate the structural motion to potential ubiquitination and explain how the linker property affecting the degradation potency. Our results show the importance of the dynamic features in protein structure and how the linker region of a PROTAC may contribute to protein motions to achieve PROTAC-mediated POI degradation.

## Results

To understand why PROTACs, although forming similar stable E3-PROTAC-POI ternary complexes, induce different degradation efficacy of POIs, we used an integrative approach that combines docking, structural alignment and atomistic MD simulations. Due to the limited availability of experimental determined CRBN-dBETx-BRD4$^{BD1}$ conformations, we initially employed protein–protein docking to generate numerous CRBN-dBETx-BRD4$^{BD1}$ ternary complexes for four degraders with different degradation efficacies (*Appendix 1—table 1* and *Figure 1—figure supplement 1*). Subsequently, we constructed degradation machinery complexes by assembling various ternary complexes into the CRL4A E3 ligase scaffold. MD simulations and subsequent post-analysis were employed to quantify protein dynamics and to identify crucial hinge regions governing the structure–dynamics–function relationship within the degradation complexes. With quantitative data, we revealed the importance of the structural dynamics of dBETx-induced motions, which arrange positions of the surface lysine residues of BRD4$^{BD1}$ and the entire degradation machinery.

### Conformational ensembles of CRBN-dBETx-BRD4$^{BD1}$ ternary complexes

An effective design of PROTACs relies on a comprehensive understanding, how a degrader, such as dBET, yields different conformations of CRBN-dBETx-BRD4$^{BD1}$ ternary complexes, ultimately contributing to degradation efficiency. To generate the conformation ensemble, we first constructed the ternary complexes through protein–protein docking using the Molecular Operating Environment (MOE) program and removed complexes with atomic clashes. *Figure 1—figure supplement 1* illustrates the docked CRBN-dBETx-BRD4$^{BD1}$ conformations, with all conformations, except CRBN-dBET57-BRD4$^{BD1}$, presenting over 160 conformations and displaying various inter-molecular orientations between CRBN and BRD4$^{BD1}$. Notably, PROTAC dBET57, characterized by a shortest linker, exhibited a constrained protein–protein orientation, resulting in only 24 distinct ternary conformations (*Figure 1—figure supplement 1C* and *Appendix 1—table 1*). To validate the modelled conformations, we superimposed the docking results of CRBN-dBET23-BRD4$^{BD1}$ and CRBN-dBET70-BRD4$^{BD1}$ ternary complexes onto available crystal structures, revealing highly similar conformations. The computed smallest C$\alpha$-root mean square deviation (RMSD) values were 1.99 Å, and 2.60 Å, respectively (*Figure 1—figure supplement 2*). These ternary ensembles, as depicted in *Figure 1—figure supplement 1* were subsequently utilized to construct degradation machinery complexes, as detailed in the Materials and methods subsection (*Figure 2*).

Furthermore, we conducted multiple MD simulations in explicit solvent for five CRBN-dBETx-BRD4$^{BD1}$ complexes, encompassing four initial conformations derived from our docking models and one from an existing crystal structure (*Table 1*, total of 15 runs for ternary complexes). Only dBET23 crystal structure is available with the PROTAC and both proteins, while the experimentally determined

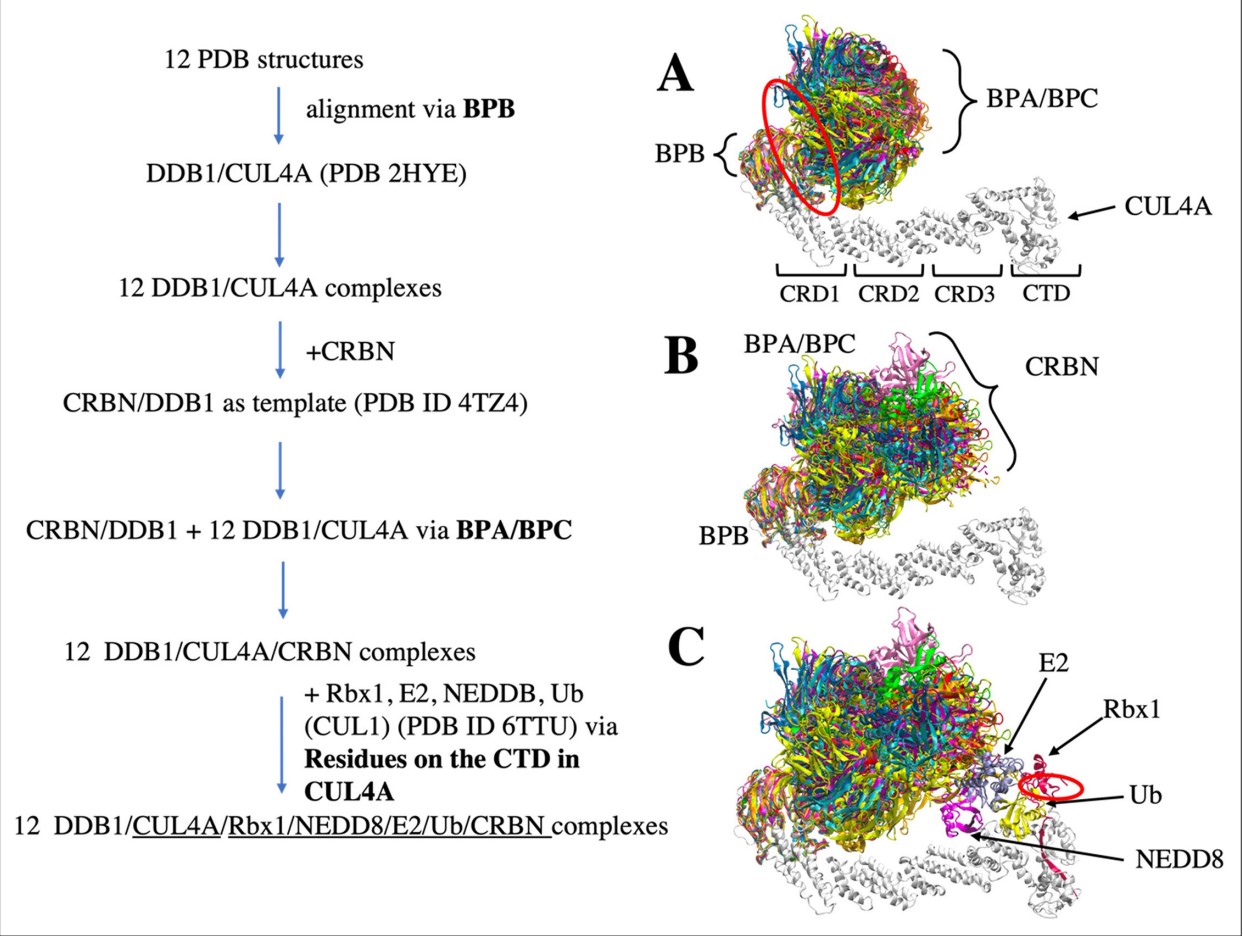

**Figure 2.** Flowchart of CRL4A E3 ligase scaffolds construction. (**A**) structure alignments of 12 DDB1 PDB structures to the CUL4A based on the BPB part. Red circle indicates a hinge loop connecting BPB and BPA/BPC in DDB1. (**B**) Structure alignment of CRBN E3 ligase to the DDB1/CUL4A based on the BPA/BPC part, resulting 12 DDB1/CUL4A/CRBN scaffolds. (**C**) E2/Rbx1/Ub/NEDD8 components (PDBID: 6TTU) are added to the /DDB1/CUL4A/CRBN scaffolds to make 12 diverse CRL4A E3 ligase scaffolds. Red circle indicates another hinge loop in the Rbx1.

The online version of this article includes the following figure supplement(s) for figure 2:

**Figure supplement 1.** Cluster C with clashes between CRBN E3 and E2.

**Figure supplement 2.** Conformational clusters of CRL4A E3 ligase scaffolds.

**Figure supplement 3.** Conformational ensemble of dBET23 from protein–protein docking.

**Figure supplement 4.** Overlapping diagram representation of common CRBN-dBETx-BRD4$^{BD1}$ ensembles found in each degradation machinery complex.

**Figure supplement 5.** Clusters of PROTAC conformations from top-ranked docking scores.

**Figure supplement 6.** Root-mean-squared deviation (RMSD) analysis for each degradation machinery complexes.

ternary complexes of dBET1, dBET57 and dBET70 are not available. During 400-ns simulations, the two warheads of a PROTAC bound tightly to CRBN and BRD4$^{BD1}$, while the linker displayed high flexibility by adopting different conformations and facilitating the interaction between the two proteins across different protein–protein contact surfaces (*Figure 3*). The residue contact map throughout the 400-ns MD simulation also showed different patterns of protein-protein interactions, indicating that the linkers were able to adopt different conformations (*Figure 3—figure supplement 1*). Interestingly, despite the molecular docking results suggested limited fluctuation in CRBN-dBET57-BRD4$^{BD1}$, our MD simulations revealed considerable variation in ternary conformations. The observed large-scale motions in all CRBN-dBETx-BRD4$^{BD1}$ complexes, as predicted by both docking and MD simulations, underscored the importance of CRBN and BRD4$^{BD1}$ orientations influenced by the presence of dBETs as key factors of contributing to degradation efficiency.

**Table 1.** List of molecular dynamics (MD) simulations for ternary complex CRBN-dBETx-BRD4$^{BD1}$ and the degradation machinery complex.

A1 and B1 indicate that the complex was assembled using scaffold cluster A1 and B1, respectively. The subscript number after each dBET degrader indicates the conformation index number from the protein–protein docking results. B1_dBET1$_{md3}$ indicates that a ternary conformation obtained from a CRBN-dBET1$_{\#35}$-BRD4$^{BD1}$ MD run (*Figure 3* color orange) was used to build the initial conformation.

| MD index | Run | MD index | Run | MD index | Run |
|---|---|---|---|---|---|
| CRBN-dBET1$_{\#35}$-BRD4$^{BD1}$ | 3 | A1_dBET1$_{\#35}$ | 1 | B1_dBET1$_{md3}$ | 2 |
| CRBN-dBET23$_{\#14}$-BRD4$^{BD1}$ | 3 | A1_dBET23$_{\#14}$ | 1 | B1_dBET23$_{\#14}$ | 2 |
| CRBN-dBET57$_{\#9}$-BRD4$^{BD1}$ | 3 | A1_dBET57$_{\#9}$ | 1 | B1_dBET57$_{\#9\_MD1}$ | 2 |
| | | | | B1_dBET57$_{\#9\_MD2}$ | 1 |
| CRBN-dBET70$_{\#91}$-BRD4$^{BD1}$ | 3 | A1_dBET70$_{\#91}$ | 1 | B1_dBET70$_{\#91}$ | 2 |
| CRBN-dBET23$_{xray}$-BRD4$^{BD1}$ | 3 | A1_dBET23$_{xray}$ | 1 | B1_dBET23$_{xray}$ | 1 |

## Modelled degradation machinery complexes

To facilitate ubiquitination, once a stable CRBN-dBETx-BRD4$^{BD1}$ complex is formed, led by CRBN, the ternary complex is assembled with an E3 ligase scaffold for ubiquitination. For this purpose, we utilized a widely employed scaffold, CRBN/DDB1/CUL4A/NEDD8/Rbx1/E2/Ub, to assemble our ternary complex, with CRBN binding to the adaptor protein DDB1 to bring the target protein into proximity with E2/Ub (*Figure 1B*). Given the dynamic nature of the scaffold complex, we selected 12 distinct DDB1 crystal structures to construct multiple scaffold conformations (*Bai et al., 2021*). Notably, among these structures, only one of the 12 PDB files had CRBN bound to DDBI (PDB ID 4TZ4). Using this crystal structure as a template, we further built 12 CRL4A E3 ligase scaffolds: DDB1/CUL4A/NEDD8/Rbx1/E2/Ub/CRBN. Subsequently, three of the resulting 12 complexes were found to contain clashes and were excluded from further analysis (*Figure 2—figure supplement 1*). The remaining nine CRL4A E3 ligase scaffolds all formed a ring-like overall shape. We further analyzed the conformations and clustered them using a distance between the E2 and CRBN interface resulting two distinct groups: the ring-forming cluster A, characterized by a gap distance ranging from ~1.0 Å to 10.0 Å (*Figure 2—figure supplement 2A*, clusters A1-5), and the ring-open cluster B, featuring a much larger gap distance ranging from ~14 Å to 35 Å (*Figure 2—figure supplement 2B*, clusters B1-4). Subsequently, we allocated 561 modeled CRBN-dBETx-BRD4$^{BD1}$ conformations, along with a crystal structure (CRBN-dBET23-BRD4$^{BD1}$), into the nine CRL4A E3 ligase scaffolds, labeled A1 to A5 and B1 to B4, resulting in the construction a total of 5058 degradation machinery complexes.

According to the mechanistic studies (*Lv et al., 2021*; *Valimberti et al., 2015*), both a Lys residue of the POI and a Gly residue in the C-terminal G-G motif in Ub must be positioned within an Asp-rich catalytic site for the ubiquitination reaction to occur, with Ub conjugated to the POI by forming a covalent bond between Lys residue and Gly residue (*Figure 1C*). Using the distance between one Lys residue of BRD4$^{BD1}$ and the C-terminal Gly residue of Ub as a criterion, we examined each Lys residue of BRD4$^{BD1}$ across the 5058 degradation machinery complexes. Any complex in which the distance between any Lys of BRD4$^{BD1}$ and the C-terminal Gly of Ub was found to be <16 Å was retained, resulting in the identification of 1226 degradation machinery complexes with diverse conformations (*Table 2*). For instance, within cluster B1 scaffold, dBET23 yielded 104 conformational degradation complexes (*Figure 2—figure supplement 3*). It is noteworthy that in each degradation machinery complex, at least one Lys residue was found within the cutoff criterion. However, due to the wide ring-open conformations of clusters B3 and B4, none of the modelled degradation complexes utilizing these two clusters satisfied our criterion. Given that a distance of 16 Å may appear to be relatively large for catalysis (*Bai et al., 2022*), we postulated that the protein dynamics could potentially bring the relevant Lys and Gly residues into close proximity to facilitate the ubiquitination reaction.

Among the docked ensembles of each CRBN-dBETx-BRD4$^{BD1}$, all top-ranked ternary complexes from protein–protein docking were successfully integrated with the scaffold clusters A and B without encountering clashes (*Figure 2—figure supplement 4*). For example, dBET1$_{\#35}$ and dBET70$_{\#91}$

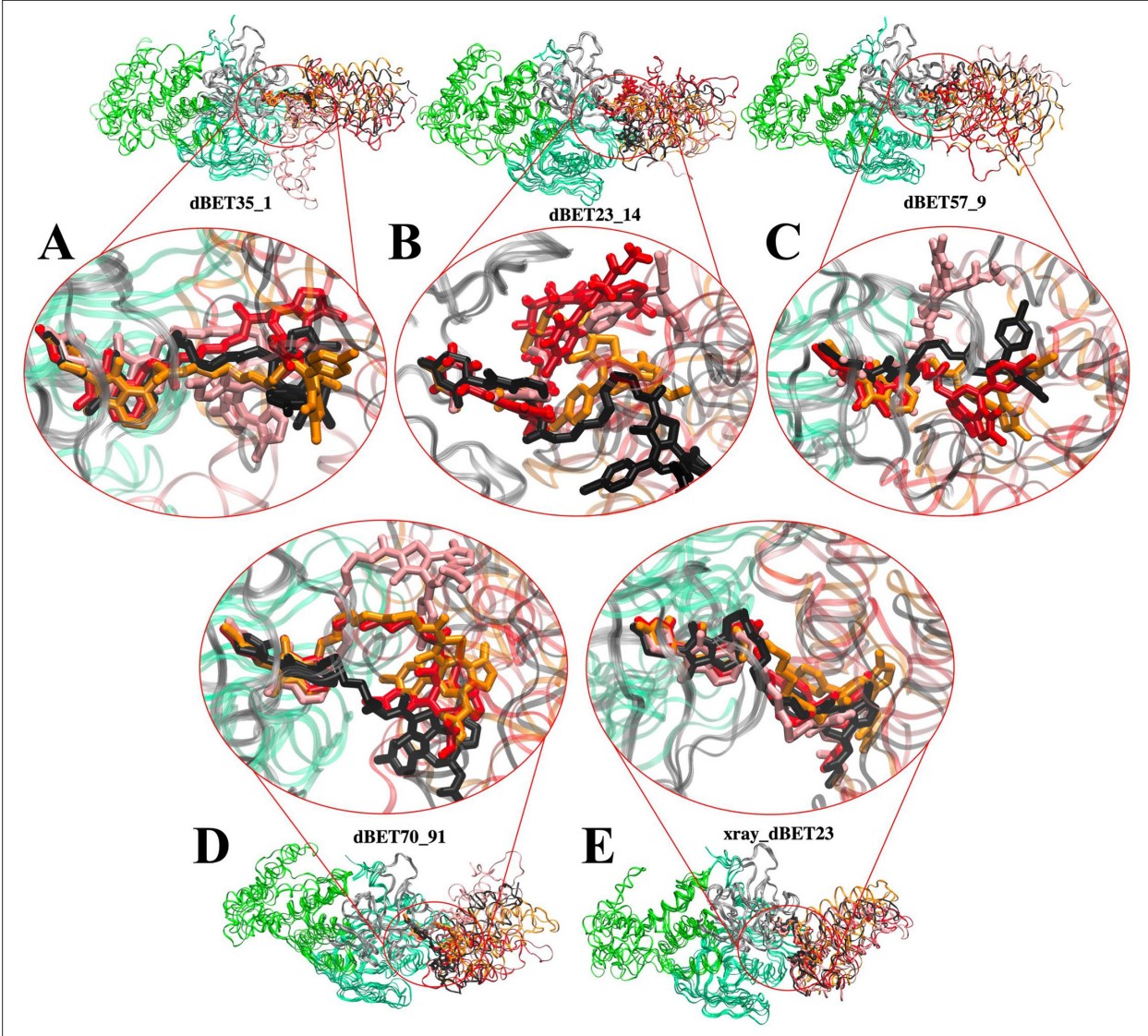

**Figure 3.** Flexibility of dBET PROTACs. The last molecular dynamics (MD) frame from three MD runs was aligned with initial frame of the C-terminus domain (gray) of CRBN. Two warheads bound tightly to the respective proteins, whereas the linker is highly flexible, adopting various conformations. MD run 1 (red), MD run 2 (pink), MD run 3 (orange), and initial frame (black). (**A**) CRBN-dBET1$_{\#35}$-BRD4$^{BD1}$. (**B**) CRBN-dBET23$_{\#14}$-BRD4$^{BD1}$. (**C**) CRBN-dBET57$_{\#9}$-BRD4$^{BD1}$. (**D**) CRBN-dBET70$_{\#91}$-BRD4$^{BD1}$. (**E**) CRBN-dBET23$_{xray}$-BRD4$^{BD}$.

The online version of this article includes the following figure supplement(s) for figure 3:

**Figure supplement 1.** Residues Contact map between PROTACs and BRD4$^{BD1}$ throughout 400ns of MD simulation.

**Figure supplement 2.** Analysis of protein-protein interaction energies of ternary complexes for each dBETs.

exhibited the most favorable protein–protein docking energies (*Appendix 1—table 2*). Intuitively, one might prioritize the ring-forming cluster A due to its smaller gap between the E2 and CRBN interface, potentially facilitating the proximity of Lys and Gly for ubiquitination. However, the conformational ensemble of each CRBN-dBETx- BRD4$^{BD1}$ provided numerous possible orientations for assembly within the degradation complex, regardless of whether the CRL4A E3 ligase scaffold adopts a ring-forming or ring-open conformation.

Moreover, when constructing a degradation complex, all the highest-ranked ternary complexes from docking scores displayed similar conformations for dBETs, implying that the preferred PROTAC conformations may be predicted. These popular ternary complexes exhibited similar protein–protein contacts and linker conformations (*Figure 2—figure supplement 5*). Our MD simulations further

**Table 2.** Number of ternary complexes used in the molecular docking and construction of degradation machinery complex.

| | | Cluster A | | | | | Cluster B | | | |
|---|---|---|---|---|---|---|---|---|---|---|
| | | Number of conformations for construction of degradation machinery complex | | | | | | | | |
| PROTAC | Conformations from protein–protein docking | A1 | A2 | A3 | A4 | A5 | B1 | B2 | B3 | B4 |
| dBET1 | 186 | 81 | 51 | 17 | 22 | 38 | 68 | 106 | 0 | 0 |
| dBET23 | 168 | 42 | 52 | 87 | 0 | 23 | 104 | 86 | 0 | 0 |
| dBET57 | 24 | 19 | 12 | 0 | 5 | 7 | 17 | 17 | 0 | 0 |
| dBET70 | 183 | 37 | 73 | 73 | 0 | 10 | 96 | 83 | 0 | 0 |

demonstrated that these initial structures used for modeling degradation complexes effectively directed BRD4$^{BD1}$ toward ubiquitination (see next subsection).

## Protein structural dynamics in degradation machinery complex

Among the 1226 distinct degradation machinery complexes constructed, none showed the structure of a Lys residue of BRD4$^{BD1}$ was positioned for ubiquitination. Therefore, for each PROTAC, we selected a degradation machinery complex conformation from clusters A and B with a top docking score CRBN-dBETx-BRD4$^{BD1}$ ternary conformation for MD simulations (*Table 1*). To further evaluate the docking scores, we also performed protein–protein interaction energy calculations using MD runs initiated by these ternary conformations with top docking scores. The energy calculations confirmed the thermodynamic stability of top-ranked ternary complexes obtained through protein–protein docking, highlighting the robust binding of their associated dBETs (*Figure 3—figure supplement 2*). Moreover, they remained structurally stable throughout the MD runs, as evidenced by RMSD plots (see RMSD plots in *Figure 2—figure supplement 6*).

After confirming that each CRBN-dBETx-BRD4$^{BD1}$ used for constructing the degradation complexes represented popular ternary conformations, we conducted classical MD simulations for these degradation complexes to observe structural dynamics and local interactions that lead to successful ubiquitination. Notably, to ensure that the compatibility of each CRBN-dBETx-BRD4$^{BD1}$ with both clusters A and B of CRL4A E3 ligase scaffolds, we also selected a few ternary complexes from our MD runs (*Table 1*). As depicted in *Figure 1C*, a successful ubiquitination reaction requires protein conformational arrangements that bring a surface Lys of BRD4$^{BD1}$ close to Gly of Ub, where at least one Asp of E2 positioned in proximity to the same Gly to facilitate charge transfer (*Appendix 1—table 3*). Therefore, we employed two criteria: (1) ensuring that the distance between Lys (N atom) and Gly (C atom) was less than 10 Å; and (2) confirming that distance between Asp (O atom) of E2 and the same Gly (O atom) was less than 6 Å to determine the likelihood of ubiquitination.

Although none of the initial modelled machinery complexes exhibited a Lys residue of BRD4$^{BD1}$ being ready for ubiquitination, our MD sampling facilitated the movement of different Lys residues of BRD4$^{BD1}$ from distance greater than 16 Å of the catalytic pocket on E2 to establish close contacts with Gly of Ub for each dBET. Specifically, we observed significant large-scale motions within degradation complex B1_dBET1$_{\#35}$ (cluster B, docked conformation #35), which recruited three Lys residues K72, K76 and K111 at distances greater than 16 Å, positioning them in the catalytic pocket on E2. Concurrently, a conserved charged residue Asp of E2 approached the C-terminus G75 of Ub, potentially facilitating isopeptide bond formation (*Figure 4A and B*). Similarly, degradation complex B1_dBET23$_{\#14}$ (cluster B, docked conformation #14) and B1_dBET70$_{\#91}$ (cluster B, docked conformation #91) also brought K99 of BRD4$^{BD1}$ into proximity of the catalytic cavity of E2, where one Asp of E2 approached the the C-terminus G75 of Ub as well (*Figure 4C, D, G and H*).

In contrast, degradation complex B1_dBET57$_{\#9\_MD1}$ (cluster B, conformation from MD run 1 initiated with docked conformation #9) positioned K102 of BRD4$^{BD1}$ in close proximity to the catalytic pocket on E2 (*Figures 4E and 3F*). Notably, BRD4$^{BD1}$ of dBET1, dBET23 and dBET70 exhibited a closer proximity to Ub (~6 Å) as compared to dBET57 (~10 Å; *Figure 4A, C, E and G*). Despite degradation complex B1_dBET1$_{\#35}$ positioned three Lys residues of BRD4$^{BD1}$ to form stable interactions with the Gly of Ub, the surrounding Asp of E2 was unstable which hindered a successful ubiquitination process (*Figure 4A*). Further analysis revealed that dBET23 (~85%) and dBET70 (~78%) exhibited a

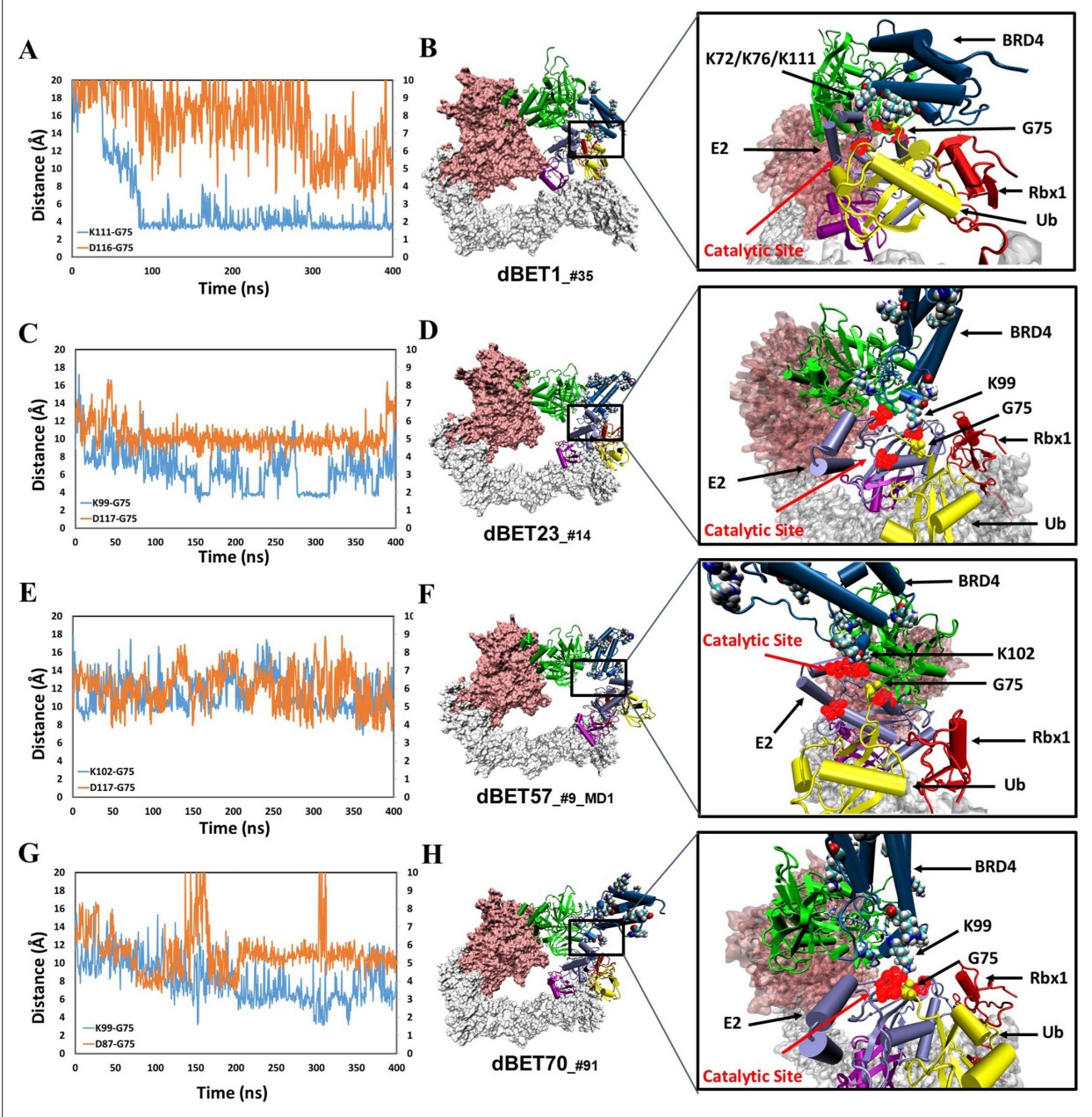

**Figure 4.** structural dynamics and local interaction of each degradation machinery. (**A**) K111 and D116 of dBET1_#35 (Distance of K72/76 is reported in *Figure 4—figure supplement 1*). The left y-axis in blue presents the distance between Lys (N atom) of BRD4^BD1 and Gly (C atom) of Ub, and the right y-axis in red presents the distance between Asp (O atom) of E2 and Gly (O atom) of Ub. The same left and right y-axis representations are used in **C**, **E** and **G**. (**B**) K72/76/111 of BRD4^BD1 reaches the negatively charged catalytic site of E2. (**C**) K99 and D117 of dBET23_#14. (**D**) K99 of BRD4^BD1 reaches the negatively charged catalytic site of E2. (**E**) K102 and D117 of dBET57_#9_MD1. (**F**) K102 of BRD4^BD1 reaches the negatively charged catalytic site of E2. (**G**) K99 and D87 of dBET70_#91. (**H**) K99 of BRD4^BD1 reaches the negatively charged catalytic site of E2. (refer to *Figure 4—figure supplement 3* for data of second seed of MD simulation).

The online version of this article includes the following figure supplement(s) for figure 4:

**Figure supplement 1.** The distance between Lys (N atom) of BRD4^BD1 and Gly (C atom) of Ub from dBET1_#35 degredation complex.

**Figure supplement 2.** Quantifying the probability of isoppetide bond formation using two criteria: (1) Lys (N atom) Gly (C atom) distance and (2) Asp (O atom) and Gly (O atom) distance.

**Figure supplement 3.** Lys (N atom) Gly (C atom) distance and 2 Asp (O atom) and Gly (O atom) distance for second seeds of MD simulation.

higher probability of fulfilling both criteria for potential chemical reactions, whereas dBET1 (~40%) and dBET57 (~22%) had a lower likelihood (*Figure 4—figure supplement 2D*). This analysis revealed how dBET23 and dBET70 achieved high degradation efficiency, while dBET1 and dBET57 induced protein motions for bond formation, without achieving all required arrangements simultaneously, resulting in lower degradation efficiency. Our findings are consistent with experimental measurements indicating that all dBETs could degrade BRD4$^{BD1}$ but with varying degradation potency. By considering the Lys–Gly and Asp-Gly distances, our simulations underscored that dBET23 and dBET70 (with DC$_{50/5h}$ values of ~5 nM and ~50 nM, respectively) are more efficient degraders compared to dBET1 and dBET57 (with DC$_{50/5h}$ values of ~500 nM).

To gain further insights into the overall motion of the degradation complexes, we employed principal component analysis (PCA) to extract essential motions within the complex; Notably, the first two PC modes, PC1 and PC2, accounted over 60% of the overall motions in most MD runs (*Appendix 1—table 4*). While the specific nature of these motions varied depending on the system, the dBETs induced an inter-protein orientation between CRBN and BRD4$^{BD1}$, with two hinge regions: loops in Rbx1 and DDB1 proteins contributing prominently to the essential motions of the complex (*Figure 5—figure supplement 1*). These two hinge regions exhibited twisted and opposite direction motions, facilitating the rearrangement and bringing BRD4$^{BD1}$ closer to the catalytic cavity of E2, which is a critical step for successful ubiquitination. Indeed, the essential motions revealed by the first two PC modes clearly revealed the plasticity between the protein–protein interfaces in the degradation machinery. Notably, the most significant motion in PC1 shifted E2/Ub/Rbx1/NEDD8 closer to BRD4$^{BD1}$ (*Figure 5—figure supplement 2*). Furthermore, the pairwise force matrix between individual residues within the degradation complex offered insights into the non-covalent interactions governing the dynamic behavior of the complex. This interaction network, starting from the dBETs' linker and extending to the hinge loop of DDB1 and Rbx1 (*Figure 5—figure supplement 3*), indicated a clear

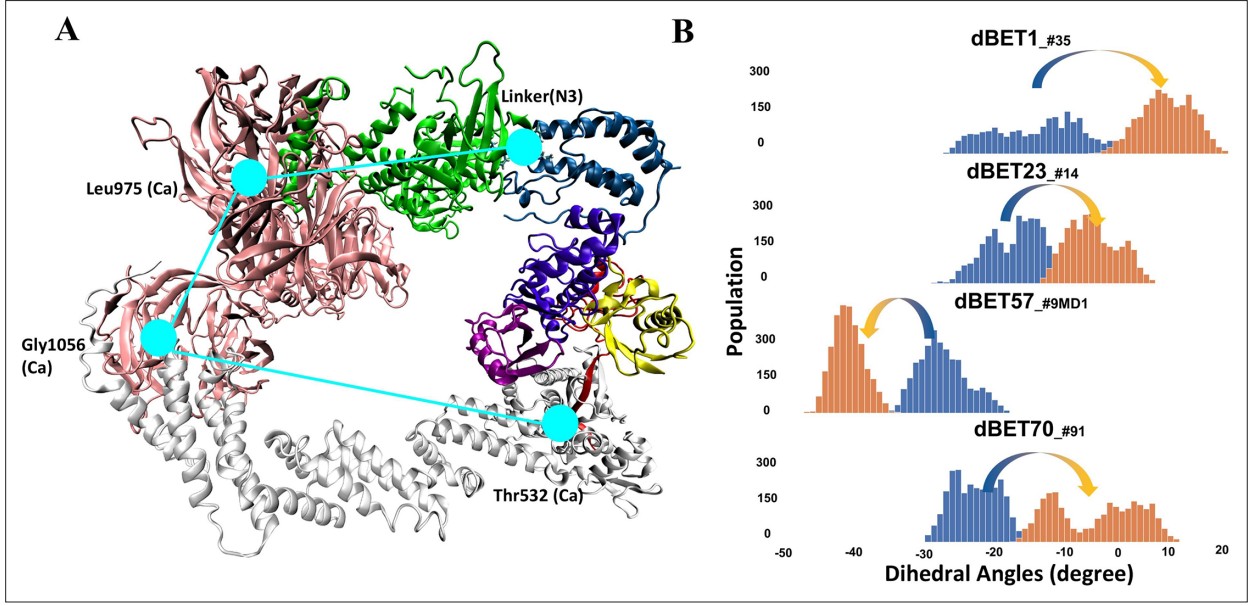

**Figure 5.** Quantifying the motion of degradation complex. (**A**) Defining the pseudo dihedral angles, Linker(N3)-Leu975(Ca)-Gly1056(Ca)-Thr532(Ca) to capture the essential motion of the degradation complex. (**B**) Dihedral angle histogram shows the population distribution of the first 100 ns (blue) and last 100 ns (orange) of the MD simulation. The distribution shows a ~10-15° dihedral angle shift.

The online version of this article includes the following figure supplement(s) for figure 5:

**Figure supplement 1.** Dynamic nature of degradation complex.

**Figure supplement 2.** First PC mode of the predicted ternary complexes.

**Figure supplement 3.** Correlation between PROTACs and hinge motion.

**Figure supplement 4.** Attractive force between Lys of BRD4$^{BD1}$ and Gly of Ub.

**Figure supplement 5.** Comparing degradation machinery complexes' dynamic of dBET57$_{\#9\_MD1}$ and dBET57$_{\#9\_MD2}$ with different BRD4$^{BD1}$ initial conformations.

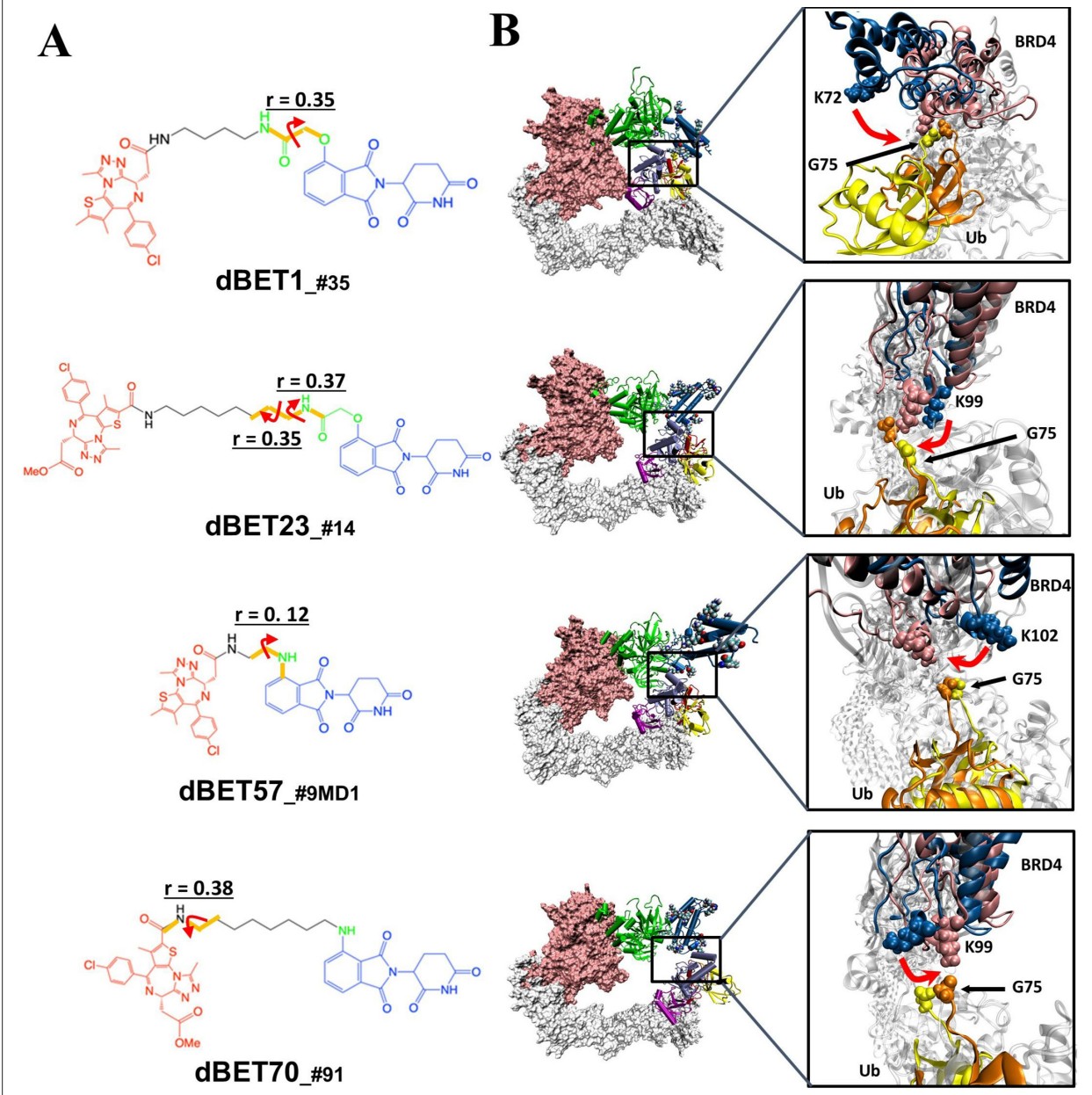

**Figure 6.** Quantification of the correlated motion between the linker and degradation complex. (**A**) Dihedral angles correlated with the degradation complex. r is the dihedral correlation coefficient (See Methods for detailed calculation). (**B**) Motion of BRD4$^{BD1}$ and Ub indicates a shift between the initial frames (BRD4$^{BD1}$, blue; Ub, yellow) and final frames (BRD4$^{BD1}$, pink; Ub, orange) of MD simulation. K72, K99 and K102 engaged in the interaction with G75 of Ub in each degradation complex, which implies their potential for degrading BRD4$^{BD1}$. For visualization purposes we present only K72 of dBET1$_{\_\#35}$, but K76 and K111 can also engage in interaction with G75 (See **Figure 4**).

correlation between linker rotation and the movement of the entire degradation complex. Our study highlights the importance of structural dynamics of the degradation complex during the ubiquitination processes, which is influenced by a dBET ligand.

To quantify the correlation between linker rotation and the movement of the degradation complex, we selected four atoms shown in **Figure 5A** to construct a pseudo dihedral angle and used histograms to depict the population of these pseudo angles. Deviations between the histograms from different time periods indicated the motion of the degradation complex during the MD simulations. Across the first and last 100 ns of an MD run, the degradation complexes large motions with the presence of each dBET (**Figure 5B**). Given that the only distinction among the four dBETs was the linker, we observed

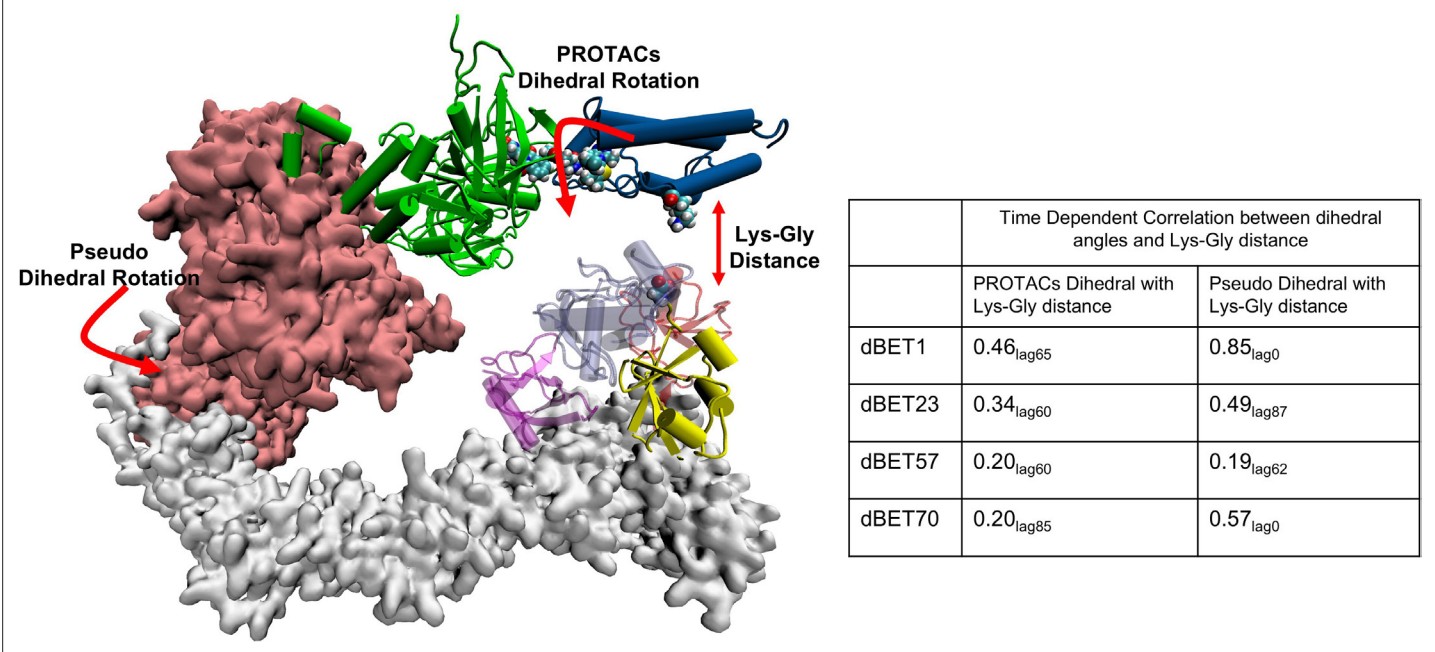

**Figure 7.** Time-dependent correlation between PROTACs/Pseudo dihedral angles and Lys-Gly interaction. The peak values of time-dependent correlation were reported. For instance, $0.46_{lag65}$ represents the correlation coefficient at 65ns of the MD simulation.

The online version of this article includes the following figure supplement(s) for figure 7:

**Figure supplement 1.** Time-dependent correlation of Psuedo dihedral angles and PROTACs dihedral angles.

that the complex motions correlated to the C-C-N3-C dihedral angles in the linkers of dBET1, dBET23, dBET70, and dBET57 (**Figure 6A**). Notably, all four dBETs exhibited substantial shifts of BRD4$^{BD1}$ of approximately 10 and 15 degrees, which underscores their flexibility in promoting extensive motion of the BRD4$^{BD1}$ protein for Lys–Gly interaction. The various motions from different degradation complexes also resulted in different sets of Lys residues potentially forming contacts with E2 and Ub. For instance, the degradation complexes dBET23$_{\#14}$ and dBET70$_{\#91}$ brought K99 of BRD4$^{BD1}$ closer to G75 of Ub (**Figure 6B**), while dBET57$_{\#9\_MD1}$ placed K102 in proximity to G75 of Ub. Despite allowing for different Lys–Gly interactions, dBET57 exhibited weaker interactions. The pairwise force analysis revealed a strong attraction between K99 and G75 of dBET23 and dBET70, and a weaker attraction between K102 and G75 of dBET57$_{\#9\_MD1}$ (**Figure 5—figure supplement 4**).

To further examine the correlation between PROTAC rotation and the Lys-Gly interaction, we performed a time-dependent correlation analysis. This analysis showed that PROTAC rotation translates motion over time, leading to the Lys-Gly interaction, with a correlation peak around 60–85 ns, marking the time of the interaction (**Figure 7** and **Figure 7—figure supplement 1**). In addition, the pseudo dihedral angles also showed a high correlation (0.85 in the case of dBET1) with Lys-Gly distance. This indicated that degradation complex undergoes structural rearrangement and drives the Lys-Gly interaction.

Given the presence of numerous local energy minima conformations in the degradation complex, MD simulations could not sample all conformations comprehensively. Furthermore, as evidenced by the protein–protein docking results (**Figure 1—figure supplement 1C**), the short linker of dBET57 yielded a less flexible ternary complex, suggesting challenges in sampling various conformations using MD runs. Our dihedral entropies analysis showed that dBET57 has ~0.3 kcal/mol lower entropies than the other dBETs with three different linkers, suggesting that dBET57 is less flexible than other PROTACs (**Figure 8**).

To address this, we selected another distinctly different ternary complex of dBET57 (**Figure 5—figure supplement 5B**) to construct another degradation complex, B1_dBET57$_{\#9\_MD2}$, for MD simulation. Unlike B1_dBET57$_{\#9\_MD1}$, B1_dBET57$_{\#9\_MD2}$ exhibited significantly more rigidity, with only minimal rotation (<5 degrees) in the pseudo dihedral angle. This rigidity limits the structural dynamics of the

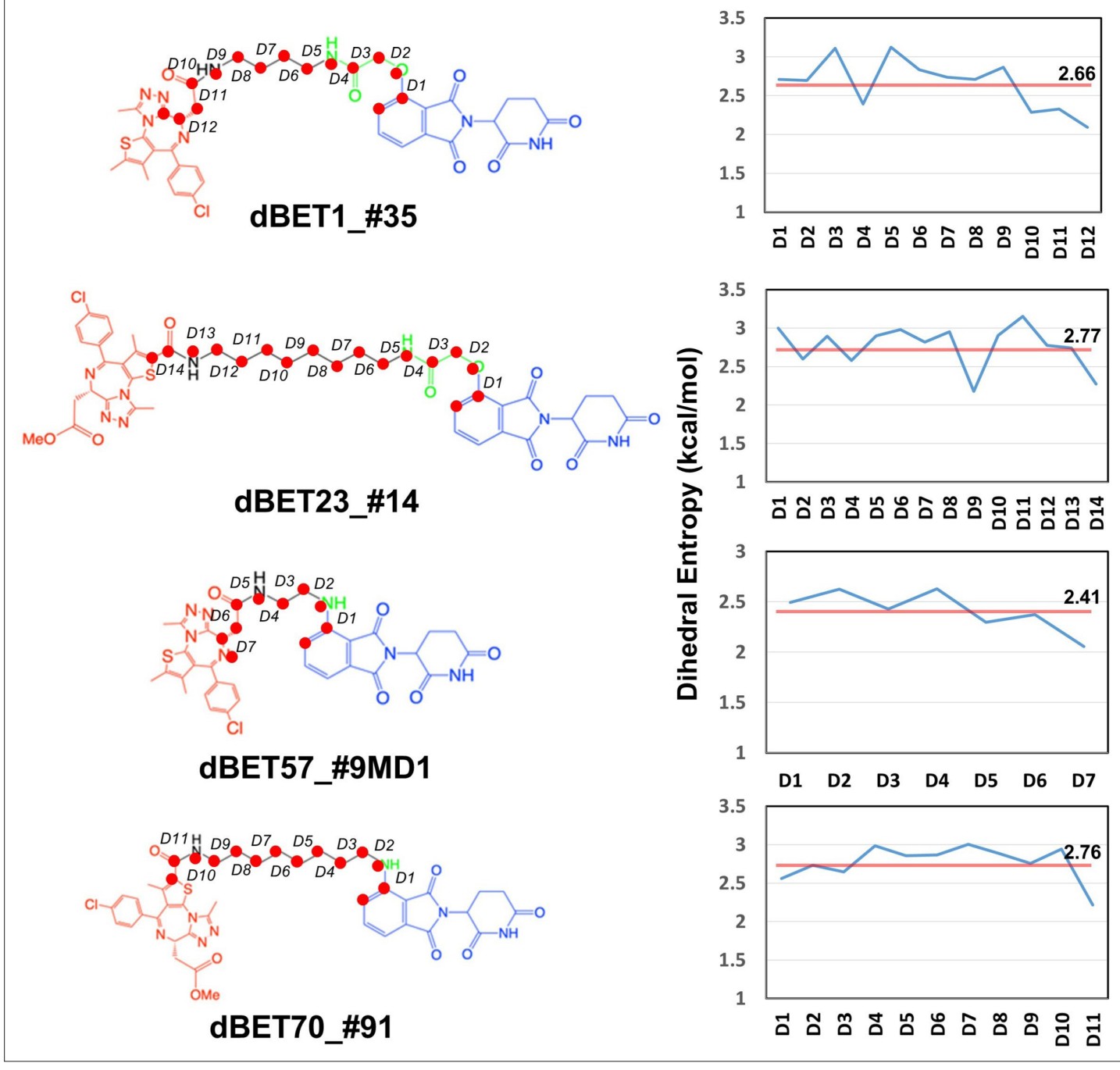

**Figure 8.** Dhedral entropies of each dihedra langles of the PROTACs using Gibbs entropy formula. The probability distribution is the probability distribution of dihedra langles. Larger entropies means the molecules is more flexible and lower values indcates a more rigid molecules.

whole complex to bring BRD4$^{BD1}$ closer to Ub for Lys–Gly interactions (*Figure 5—figure supplement 5*). This observation underscores the crucial role of the linker in guiding the motion of the degradation complex, thereby influencing its capacity for ubiquitination. In summary, the linker induces conformations of ternary complexes but also guides the motion of the degradation complex, which allows an effective Lys–Gly interaction for ubiquitination. Our studies highlight the critical role of linker flexibility in facilitating protein structural dynamics necessary for ubiquitination.

## Discussion

As bifunctional small-molecule ligands, PROTACs play a pivotal role in recruiting a neo-substrate of E3 ligase, namely the target protein POI, to form a ternary E3-PROTAC-POI complex by stabilizing the protein–protein interactions. By forming an E3-PROTAC-POI complex, E3 ligase brings the POI close to an E2 enzyme for protein ubiquitination, with both E2 and E3 proteins belonging to a larger Ub ligase machinery complex. While forming a stable E3-PROTAC-POI complex represents a critical step in ubiquitination, it alone is insufficient for subsequent ubiquitination and degradation by proteasomes. Several studies suggest that positioning the POI's Lys residues close to the E2 for ubiquitination is crucial for successful POI degradation (*Smith et al., 2019*; *Dixon et al., 2022*; *Petzold et al., 2016*). However, due to the lack of experimental determined structures for POI-bound degradation complexes, directly observing the proximity of Ub to POI's Lys residues remains challenging. In addition, proteins exhibit dynamic behaviors, and the fluctuation in the ternary E3-PROTAC-POI complex and the degradation machine can either promote or impede catalytic activity. To investigate the degradability of PROTACs, we employed a strategy combining protein docking, MD simulations, and structural and energy analysis tools. Our focus was on a series of dBET degraders, differing in their linkers, which all induced stable CRBN-dBETx-BRD4$^{BD1}$ complexes different degradation efficiencies.

One straightforward approach to promote ubiquitination is to obtain a stable E3-PROTAC-POI ternary complex to allow pre-organized orientation placing a Lys residue(s) of the POI close to the E2/Ub catalytic site in the degradation machine. However, experimental determined structures for degradation complexes are scarce. Protein–protein docking can reveal low energy ternary complexes and popular conformational ensembles, providing insights into potential orientations for ubiquitination. Although protein–protein docking is usually quick (i.e. <50 hr to sample 5000 protein–protein complexes using one CPU in computation), the docked E3-PROTAC-POI conformations may not be the functionally active form to induce successful ubiquitination. Nevertheless, results from protein–protein docking are highly informative and suggest potential flexibility of a ternary complex, as shown in *Figure 1—figure supplement 1*. However, MD simulations or other conformational search methods can further generate more stable ternary conformations, and the sampling from protein–protein docking may not be thorough. As illustrated by dBET57, protein–protein docking generated only similar ternary complex conformations (*Figure 1—figure supplement 1C*). In contrast, classical MD simulations using all atoms and explicit water molecules sampled additional low energy conformations (*Figure 3C*).

Using the protein–protein docking results, multiple thermodynamically stable ternary E3-PROTAC-POI conformations were placed in the degradation machine to access whether surface Lys residues of POI were appropriately oriented for ubiquitination. However, relying solely on a handful of static conformations for predicting POI degradability is not insufficient. For example, our docking results showed that none of the Lys residues of BRD4$^{BD1}$ was ready for ubiquitination. Notably, our modelled CRL4A E3 ligase scaffolds in *Figure 2—figure supplement 2* show two distinct conformations: a ring-forming conformation (cluster A) with a gap distance approximately 1.0 Å to 10.0 Å, and a ring-open conformation (cluster B) with a much larger gap distance, approximately 14 Å to 35 Å. When docking the ternary complex into the ring-forming scaffold (*Figure 2—figure supplement 2A*), the orientations of BRD4$^{BD1}$ tended to be sterically confined, limiting their range of motion to bring one Lys residue of BRD4$^{BD1}$ closer to the Gly of Ub. In contrast, using MD to simulate protein dynamics, CRL4A E3 ligase scaffolds with ring-open conformations provided a larger space that may allow an assembled BRD4$^{BD1}$ to properly rearrange its conformations for successful ubiquitination, even though the initial distance between BRD4$^{BD1}$ and Ub is far. Our studies demonstrated that using CRBN-dBETs-BRD4$^{BD1}$ conformations sampled by protein–protein docking and/or MD simulations provided reasonably good initial structures to model the structural dynamics of the degradation machinery. The essential motions were captured by PCA, which illustrated that all dBETs studied can produce motions to bring BRD4$^{BD1}$ closer to Ub, thus supporting their degradability. Atomistic simulations revealed that specific Lys residue(s) of BRD4$^{BD1}$ (i.e. K99, K102 and K72/76/111 of dBET23 and dBET70, dBET57 and dBET1, respectively) were attracted by polar residues around Ub and E2.

The analysis of how a PROTAC, particularly the linker region, affects the overall dynamics and specific arrangements to bring Lys close to the catalytic site offers further information on linker design. The slight variations in linker length and composition that may significantly alter the degradability of a PROTAC was puzzling. Even though PROTACs can strengthen interactions between E3 and POI,

protein degradation is not always associated with forming stable ternary complexes. One hypothesis is that some ternary complexes have a preferred E3/POI orientation in the degradation machinery complex to result in efficient ubiquitination. Here, we demonstrated that by using accurately modeled initial conformations, performing a few hundred nanosecond classical MD simulations could illustrate how protein structural dynamics orient surface Lys residues of BRD4$^{BD1}$ and Asp of E2 close to the Gly of Ub, thus offering information for predicting protein degradation. We examined the hinges and used pseudo-dihedral angles of the degradation complexes to describe the large-scale swinging motions near the end of the MD runs (300–400 ns). To understand the role of the linker of different dBETs, we compared conformation ensembles from the first and last 100 ns of MD simulations. The movement of the degradation machinery correlated with rotations of specific dihedrals of the linker region in dBETs (*Figure 6*). In addition, the, dihedral rotation of PROTACs shows correlation with the Lys-Gly interaction (*Figure 7* and *Figure 7—figure supplement 1*). This phenomenon shows that dBETs contribute to the structural dynamics of degradation complexes and suggests that linkers with different properties may lead to various large-scale motions.

The dynamics of the degradation machinery, induced by PROTACs, has implications in linker design and degradation prediction. Because the 'warhead' (thieno-triazolo-1,4-diazepine [JQ1]) used to bind to an E3 ligase and the 'warhead' (pomalidomide) used to bind to a POI are tightly bound ligands to each protein, the two warheads typically have limited motions when they are in the binding site of E3 or POI. Clearly, the length and composition of the linker are critical in degradation efficiency, and sites of conjugation to each warhead also affect the stability of the ternary complexes. Despite no general rules that apply to every system for PROTAC linker design, guidelines used in designing small-molecule drugs binding to their target proteins still hold. First, the linker should have sufficient length to form a ternary complex. Although longer and more flexible linkers typically provide enhanced opportunities for E3-PROTAC-POI complexes to orient suitable conformations for ubiquitination, shorter and/or rigid linkers may provide a preorganized E3-PROTAC-POI for efficient catalysis. However, knowing in advance the best related orientations is challenging, but the shorter linker in our systems, dBET57, had disadvantages in accommodating a Lys residue in the E2/Ub catalytic region for forming stable interactions for catalysis. In contrast, dBET23 and dBET70, with greater flexibility, allowed BRD4$^{BD1}$ to adopt diverse conformations for catalysis, ultimately leading to more efficient degradation. Notably, dBET1 recruited multiple Lys residues in the E2/Ub catalytic region and K111 was highly stable. However, the unstable Asp from the surrounding prevented charge transfer, and thus hindered the isopeptide bond formation for degradation. Notably, dBET1 and dBET57 still induce stable CRBN-dBET57-BRD4$^{BD1}$ and CRBN-dBET1-BRD4$^{BD1}$ complexes, which effectively assemble the neo-substrate BRD4$^{BD1}$ with the degradation machine. However, the either Lys–Gly or Asp-Gly attractive force was unstable, which reduced the chances for successful ubiquitination.

Revealing structural dynamics of the entire degradation complex that bring surface Lys residue(s) of BRD4$^{BD1}$ for target ubiquitination explains the degradability of these dBETs. Notably, the classical MD trajectories with atomistic details offer information about overall protein motions and specific Lys residues critical for the degradability of BRD4$^{BD1}$ for further investigation. Theoretically speaking, excessively long MD simulations should be able to sample a wide range of protein conformations. However, long MD simulations can still stick in local minima and are computationally expensive. We demonstrated the use of the most stable ternary CRBN-dBETx-BRD4$^{BD1}$ conformations from protein–protein docking to facilitate sampling the correct orientation of the surface Lys residue(s) of BRD4$^{BD1}$ for ubiquitination. Of note, CRBN-dBET57-BRD4$^{BD1}$ formed by the more rigid linker of dBET57 could be more easily trapped in a local energy minimum, and we need to select conformations from both protein-docking (dBET57$_{#9}$) and MD-sampling results (CRBN-dBET57$_{#9}$-BRD4$^{BD1}$) to construct the degradation machinery complex for structural dynamics studies. Different E3 ligases may also result in significantly different related orientation when bound to the same POI. Although static conformations from experimentally determined structures may suggest whether a dBET orients CRBN-dBETx-BRD4$^{BD1}$ conformations suitable for degradation, real motions cannot be predicted directly from static conformations. Examining the dynamic nature of the degradation complex provides a more complete picture of the conformations of both ternary CRBN-dBETx-BRD4$^{BD1}$ and machinery complexes that determine Lys accessibility for POI ubiquitination, an important factor for consideration when optimizing a PROTAC to improve its potency.

## Materials and methods

### Modeling CRBN-dBETx-BRD4$^{BD1}$ ternary complexes

The inhibitor (JQ1)-bound BRD4$^{BD1}$ crystal structure (PDB ID: 3MXF) *Filippakopoulos et al., 2010* and pomalidomide-bound CRBN crystal structure (PDB ID: 4CI3) *Fischer et al., 2014* were downloaded from the PDB website. Four BRD4$^{BD1}$ PROTACs (dBET1, dBET23, dBET57 and dBET70) were prepared using the builder program in the Molecular Operating Environment (MOE). Ternary complex ensembles were generated using the method 4B protocol implemented in the MOE program (Chemical Computing Group; *Drummond et al., 2020*). CRBN-dBETx-BRD4$^{BD1}$ ternary complex was modelled in four steps: (1) CRBN-BRD4$^{BD1}$ protein-protein docking with a conventional global protein-protein docking approach; (2) Robust conformational sampling of isolated dBET; (3) CRBN-BRD4$^{BD1}$ pre-generated poses and dBET conformation alignment with respective inhibitors in either protein kept intact while using Maximum Common Substructure (MCS) approach to determine the match on-the-fly between the binding ligands in the protein-protein docking ensemble and supplied dBET; (4) scoring and clustering of modelled CRBN-dBETx-BRD4$^{BD1}$ ternary complex. Before performing protein–protein docking in step 1, two crystal structures were prepared by adding missing hydrogens and assigning an appropriate protonation state for each protein. In the protein–protein docking step 1, two binary complexes interacted without a linker connecting two warheads. The protein–protein poses were generated, and conformations were stored for the next step. In the dBET (or PROTAC in general) conformational sampling step 2, five dBETs were provided for robust conformational searching. Each iteration was set to 10,000. Ligand core root mean square deviation (RMSD) was set as the default. In the dBETx conformational searching process, each binding warhead in the dBETs was held rigid to retain its bound conformation, whereas the linker region was simulated on-the-fly. A MCS approach was applied in the program to maximize dBET warheads alignment with the binding warheads. The simulated dBET conformations were then stored for the next step. In the protein–protein pose and dBET conformation alignment step, protein–protein poses with two warheads in each binding pocket were aligned with the dBETs' two binding moieties. The generated ternary complex CRBN-dBETx-BRD4$^{BD1}$ ensembles were minimized, scored, clustered. Selected represented conformations with docking scores were stored in pdb format for further visualization analysis.

### Constructing CRL4A E3 ligase scaffold: DDB1/CUL4A/NEDD8/Rbx1/E2/Ub/CRBN

We obtained 12 DNA damage-binding protein 1 (DDB1) PDB structures from the PDB website and named them by the index ID: A1_DDB1 (PDB ID: 4E54), A2_DDB1 (PDB ID: 3E0C), A3_DDB1 (PDB ID: 3I8E), A4_DDB1 (PDB ID: 4TZ4), A5_DDB1 (PDB ID: 4A0L), B1_DDB1 (PDB ID: 2B5L), B2_DDB1 (PDB ID: 3EI4), B3_DDB1 (PDB ID: 3EI3), B4_DDB1 (PDB ID: 6PAI), C1_DDB1 (PDB ID: 6FCV), C2_DDB1 (PDB ID: 4A08), and C3_DDB1 (PDB ID: 4A0B). CUL4A with its DDB1 crystal structure was downloaded (PDB ID: 2HYE) and used as a template. 12 DDB1 crystal structures were superimposed on the CUL4A/DDB1 template via β-propeller domains (BPB) to build 12 DDB1/CUL4A structures. CRBN/DDB1 from A4_DDB1 (PDB ID: 4TZ4) was used as a template, and the rest of the DDB1/CUL4A structures were superimposed via the BPA/BPC domain to build 12 DDB1/CUL4A/CRBN complexes. Next, E2D1 (PDB ID 5FER chain B) was aligned to (PDB ID 6TTU) to replace E2D2. We then aligned the 12 CRBN/DDB1/CUL4A complexes via residues in the C-terminal domain (residues from 416 to 672) in the CUL4A. Components having NEDD8, Rbx1, E2 and Ub with CUL1 (PDB ID 6TTU) were added to the above 12 CRBN/DDB1/CUL4A structures. The final 12 DDB1/CUL4A/Rbx1/NEDD8/E2/Ub/CRBN E3 ligase scaffolds were ready for the next step (*Figure 2* for flowchart). Next, we analyzed and classified the 12 CRL4A E3 ligase scaffolds DDB1/CUL4A/NEDD8/Rbx1/E2/Ub/CRBN based on the interface distance between the CRBN E3 ligase and the E2. Specifically, we measured residue distances from CRBN E3 to the E2 interface plane among these 12 CRL4A E3 ligase scaffolds. If the CRBN E3 ligase residue distances to E2 plane were <10 Å, with no overlapping or clashing, we grouped those CRL4A E3 ligase scaffolds as cluster A; if the CRBN E3 ligase residue distances to E2 plane were >10 Å (the exact gap distance was from ~14 Å to ~34 Å), we grouped those scaffolds as cluster B. If CRBN E3 and E2 overlapped, we grouped those scaffolds as cluster C, considering them as clashing and abandoned them (*Figure 2—figure supplement 1*).

## Degradation machinery complex assembling: DDB1/CUL4A/NEDD8/Rbx1/E2/Ub/CRBN-dBETx-BRD4$^{BD1}$

CRBN-dBETx-BRD4$^{BD1}$ conformations generated from protein–protein docking were aligned with the CRL4A E3 ligase scaffolds using clusters A and B (total of nine CRL4A E3 ligase scaffolds) via the common CRBN structure. Because there are 12 lysine residues on the surface of BRD4$^{BD1}$ (*Figure 1— figure supplement 1A*), we considered all the lysine residues that potentially lead to ubiquitination by measuring the distance between the oxygen from the C-terminal glycine of the ubiquitin and the nitrogen atom of surface-exposed lysine residue(s) of the BRD4$^{BD1}$ by using VMD (*Winter et al., 2015*; *Appendix 1—table 1*). Among the modelled CRBN-dBETx-BRD4$^{BD1}$ conformations each with CRL4A E3 ligase scaffolds (A1, A2, A3, A4, A5, B1, B2, B3, B4), only those with the measured distance less than or close to 16 Å without clashing were kept for further study, as suggested by previous study (*Liu et al., 2022*).

## MD simulations

The top selected CRBN-dBETx-BRD4$^{BD1}$ conformations from combining protein–protein docking, and structural based steps were used for MD simulations to capture the dynamics of CRBN-dBETx-BRD4$^{BD1}$ and degradation machinery complexes (*Appendix 1—table 1*). For each dBET, we selected one CRBN-dBETx-BRD4$^{BD1}$ conformation. For reference, we also investigated and performed MD simulations for resolved crystal structures. Among four studied dBETs in this work, only CRBN-dBET23-BRD4$^{BD1}$ (PDB ID 6BN7) has PDB structures for starting the MD simulations. Other ternary complexes missing the CRBN E3 component (PDB ID 4ZC9, no dBET1 *Winter et al., 2015* or the dBET degrader (PDB ID 6BNB, no dBET57, PDB ID 6BN9 no dBET70) were not considered. Before proceeding to MD simulations, the missing part in each protein component of each CRL4A E3 ligase complex was built by using structural alignment or the homology modeling method (*Waterhouse et al., 2018*). In brief, we used (PDB ID 4A0L) as a template for A1_DDB1, (PDB ID 3L7K) as a template for A3_DDB1, and (PDB ID 5HTB) as a template for A5_DDB1 in homology modeling. For the remaining CRL4A E3 ligase complexes (A2, A4, B1, B2, B3, B4), we built the missing part directly using the homology modeling method. We chose one CRL4A E3 ligase complex from each category: A1 in category A and B1 in category B based on the high populations in that complex. Then two criteria were considered to select the top CRBN-dBETx-BRD4$^{BD1}$ for the ternary complex MD simulations and for degradation machinery complex MD simulations: 1) the common CRBN-dBETx-BRD4$^{BD1}$ shared in the category (*Figure 1— figure supplement 1* and *Figure 2—figure supplement 1*) the best protein–protein docking score (*Appendix 1—table 2*).

MD simulations were performed using the AMBER20 software package (*Hung et al., 2023*) with AMBER ff14SB force field *Maier et al., 2015* for proteins and GAFF2 *He et al., 2020* for ligand parameterization. The atomic charges were computed using AM1-BCC (*Jakalian et al., 2002*). A standard AMBER simulation protocol was used. The sequential minimization steps were performed for the hydrogen atoms, side chains and the entire protein complex to clean any clashes. The system was solvated in a TIP3P water box *Jorgensen et al., 1983* with distance 12 Å from the edge of the protein. To neutralize the CRL4A E3 ligase complex system, 40 and 46 Na$^+$ ions were added to the water box for C4 cluster and for C12, respectively, with no counter ion added to the ternary complex. Subsequently, the whole complex system with water was minimized by only focusing on water molecules, then the entire complex system. Heating of the complex system from 50 K to 300 K with increments of 25 K was achieved in 200-ps NPT simulation (time step of 2 fs) for each temperature. The system was subsequently equilibrated at 300 K for 500-ps NPT simulation with all atoms relaxed. Long-range electrostatic interactions were computed by the particle mesh Ewald method (PME; *Essmann et al., 1995*). The SHAKE constraint was applied to all atoms including hydrogens (*Ryckaert et al., 1977*). The Langevin thermostat with a damping constant of 2 ps$^{-1}$ was used to maintain a temperature of 300 K. The NPT ensemble at 300 K was used for MD production run for 400 ns, and each frame was collected at 1-ps time intervals. We generated eight CRBN-dBETx-BRD4$^{BD1}$ complex structures and performed 3 MD runs with different initial velocities for each structure. Sixteen degradation machinery complex structures were constructed, and we ran one MD for each complex (*Table 2*).

## Dihedral angle correlation

We used the T-analyst program *Ai et al., 2010* to calculate the pairwise correlations between the pseudo dihedral angles and the PROTAC dihedral angles. Each side-chain dihedral angle was recorded every 50 ps through 400 ns trajectories to generate 8000 different angles per dihedral selection. Pairwise correlations were computed using a Pearson correlation formula. We converted the side-chain dihedral angles to Cartesian coordinates by means of *equations (1)-(3)* to accurately capture their differences and means, thereby preventing erroneous computation of their correlation at the discontinuity margin (±180° or 360°/0°; *Ai et al., 2010*; *Tang and Chang, 2017*). Notably, a positive correlation between two sidechains indicates that the two sides rotate similarly during MD simulation.

$$r_{xy} = \frac{\sum_{i=1}^{n} (x_i - \bar{x}) (y_i - \bar{y})}{\sqrt{\sum_{i=1}^{n} (x_i - \bar{x})^2} \sqrt{\sum_{i=1}^{n} (y_i - \bar{y})^2}} \tag{1}$$

$$\bar{x} = arctan \left( \frac{sin(x_1) + sin(x_2) + \ldots + sin(x_n)}{cos(x_1) + cos(x_2) + \ldots + cos(x_n)} \right) \tag{2}$$

$$x_i - \bar{x} = arctan \left( \frac{sin(x_i) cos(\bar{x}) - sin(\bar{x}) cos(x_i)}{cos(x_i) cos(\bar{x}) + sin(x_i) sin(\bar{x})} \right) \tag{3}$$

$r_{xy}$ = Dihedral Pearson Correlation, $\bar{x}$ = mean of dihedral angles,

$x_i$ = side chain dihedral angles

## Interaction energy calculations

We used MM/PBSA method from AMBER20 *Miller et al., 2012* to evaluate the stability of ternary complex by computing the interaction energy between protein (CRBN-BRD4$^{BD1}$) and ligand (dBETs). From a total of 8000 MD frames making up the 400-ns ternary complex trajectories, system conformations were analyzed every 4 ns. This method computes the energy (E) of a system from the protein (CRBN-BRD4$^{BD1}$), ligand (dBETs) and protein–ligand complex (CRBN–dBETs–BRD4$^{BD1}$), and computes the interaction energy as:

$$\Delta E = E_{CRBN-dBETs-BRD4^{BD1}} - E_{CRBN-BRD4^{BD1}} - E_{dBETs} \tag{4}$$

The solute dielectric is set to 15.0 to consider the polar surface of protein and solvent dielectric was set to 80.0.

## Pairwise interaction network

Gromacs force distribution analysis (FDA) was used to compute nonbonding forces with Leonard-Jones' potential and Coulomb potential (*Stacklies et al., 2011*).The parameter 10 Å was used for the short-range interaction cutoff. The long-range electrostatic forces were computed with PME (*Essmann et al., 1995*).The first 40 ns of MD simulation were treated as equilibrium plus, and thus the FDA was performed with the following 360 ns. We calculated the sum of pair-wise forces of the degradation complex by using our in-house script. In addition, we constructed the pair-wise forces network of the intramolecular attraction of degradation complex to visualize the shortest path of the PROTAC-guided essential motion at the hinge region by using python library Networkx (*Swart et al., 2008*; *Figure 5—figure supplement 2*). To reduce the complexity of the network, we eliminated pair-wise forces that had minor attraction or repulsion. Specifically, pair-wise forces that were within –10 and 10 pN were eliminated.

## Principle component analysis (PCA)

To observe major protein motions, we performed PCA using CPPTRAJ and our in-house code (*Hotelling, 1933*; *Roe and Cheatham, 2013*). PCA of MD trajectories for all the simulated systems involved using backbone atoms in the full degradation machinery complex. The first and second PCs were analyzed to reveal the dominant motions (*Figure 5—figure supplement 1*).

# Acknowledgements

We thank Drs. Eric Fischer and Radosław Nowak for helpful discussions and for providing experimental data for protein binding and degradation efficiency, and Dr. Kevin Kou for helpful discussions regarding the mechanisms of isopeptide bond formation. This publication was made possible by Grant No. GM109045 and GM151651 from the National Institute of General Medical Sciences (NIGMS) of the NIH.

## Additional information

### Funding

| Funder | Grant reference number | Author |
|---|---|---|
| National Institute of General Medical Sciences | GM109045 | Chia-en A Chang |
| National Institute of General Medical Sciences | GM151651 | Chia-en A Chang |

The funders had no role in study design, data collection and interpretation, or the decision to submit the work for publication.

### Author contributions

Kingsley Y Wu, Ta I Hung, Conceptualization, Data curation, Software, Formal analysis, Validation, Investigation, Visualization, Methodology, Writing – original draft, Writing – review and editing; Chia-en A Chang, Conceptualization, Resources, Software, Supervision, Funding acquisition, Validation, Investigation, Methodology, Writing – original draft, Project administration, Writing – review and editing

### Author ORCIDs

Kingsley Y Wu ⬤ http://orcid.org/0000-0002-7989-1584
Ta I Hung ⬤ https://orcid.org/0009-0008-9702-3829
Chia-en A Chang ⬤ https://orcid.org/0000-0002-6504-8529

Reviewer #1 (Public review): https://doi.org/10.7554/eLife.101127.3.sa1
Reviewer #2 (Public review): https://doi.org/10.7554/eLife.101127.3.sa2
Reviewer #3 (Public review): https://doi.org/10.7554/eLife.101127.3.sa3
Author response https://doi.org/10.7554/eLife.101127.3.sa4

## Additional files

### Supplementary files

MDAR checklist

### Data availability

The PDB format of the four modeled degradation complex structures are provided in supplementary files. Other input/output and trajectory files and in-house codes for data analysis are available on Zenodo.

The following dataset was generated:

| Author(s) | Year | Dataset title | Dataset URL | Database and Identifier |
|---|---|---|---|---|
| Ta H | 2025 | PROTAC-induced Protein Structural Dynamics in Targeted Protein Degradation | https://doi.org/10.5281/zenodo.14791107 | Zenodo, 10.5281/zenodo.14791107 |

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

# Appendix 1

**Appendix 1—table 1.** Summary of the predicted CRBN-dBETx-BRD4$^{BD1}$ ensembles from protein-protein docking.

Total ternary conformations were recorded, and conformations with clashes were carefully examined and manually removed.

| PROTAC | DC$_{50/5h}$ (nM)6 | Degrader Conf. | Total ternary Conf. predicted | Selected reasonable ternary complexes | Reasonable ternary complexes (%) |
|---|---|---|---|---|---|
| dBET1 | ~500 | 125 | 297 | 186 | 62.3 |
| dBET23 | ~50 | 212 | 250 | 168 | 67.2 |
| dBET57 | ~500 | 25 | 40 | 24 | 60.0 |
| dBET70 | 5 | 143 | 217 | 183 | 84.4 |

**Appendix 1—table 2.** Binding scores predicted by MOE protein-protein docking for the CRBN-dBETx-BRD4$^{BD1}$ ternary complexes.

| dBET1 Conf. index # | Etotal (kcal/mol) | dBET23 Conf. index # | Etotal (kcal/mol) | dBET57 Conf. index # | Etotal (kcal/mol) | dBET70 Conf. index # | Etotal (kcal/mol) |
|---|---|---|---|---|---|---|---|
| 35 | −7353.1 | 206 | −7272.9 | 22 | −7265.3 | 91 | −7330.9 |
| 10 | −7352.0 | 14 | −7221.1 | 24 | −7265.2 | 85 | −7326.2 |
| 27 | −7349.2 | 165 | −7193.7 | 26 | −7264.9 | 90 | −7323.8 |
| 30 | −7345.6 | 168 | −7183.9 | 9 | −7237.8 | 117 | −7319.3 |
| 101 | −7304.9 | 166 | −7177.8 | 23 | −7225.3 | 88 | −7314.0 |
| 110 | −7279.3 | 217 | −7141.8 | 1 | −7216.9 | 118 | −7308.2 |
| 36 | −7255.5 | 167 | −7141.4 | 7 | −7201.4 | 120 | −7303.2 |
| 106 | −7236.1 | 16 | −7045.1 | 4 | −7153.0 | 89 | −7297.7 |
| 114 | −7260.9 | | | 2 | −7145.9 | 119 | −7295.4 |
| 119 | −7142.3 | | | 3 | −7130.2 | 122 | −7288.8 |
| | | | | 6 | −7125.6 | 125 | −7256.5 |
| | | | | 11 | −7065.6 | 123 | −7254.4 |
| | | | | | | 98 | −7244.7 |
| | | | | | | 121 | −7237.9 |
| | | | | | | 87 | −7219.4 |
| | | | | | | 95 | −7214.7 |
| | | | | | | 96 | −7195.2 |
| | | | | | | 124 | −7104.8 |
| | | | | | | 126 | −7040.4 |

**Appendix 1—table 3.** Accessible lysine residues identified in the CRL4A E3 ligase scaffolds for each PROTAC in each cluster.

| PROTAC | A1 | A2 | A3 | A4 | A5 |
|---|---|---|---|---|---|
| dBET1 | 55, 57, 102, 112 | 55, 57, 12 | 112, 155 | 112 | 55, 57, 72, 91, 111, 112, 155, 160 |
| dBET23 | 55, 102, 141, 155, 160 | 55 ,99, 102, 112, 141,150, 160 | 72, 76, 111, 155 | N/A | 155, 160 |

*Appendix 1—table 3 Continued on next page*

*Appendix 1—table 3 Continued*

| PROTAC | A1 | A2 | A3 | A4 | A5 |
|---|---|---|---|---|---|
| dBET57 | 55, 112 | 55, 112 | N/A | 111, 112 | 111, 112 |
| dBET70 | 55, 141, 160 | 55, 99, 102, 112, 141, 150, 160 | 76, 155, 160 | N/A | 155, 160 |

| PROTAC | B1 | B2 | B3 | B4 |
|---|---|---|---|---|
| dBET1 | 55, 57, 72, 102, 111, 112 | 55, 57, 72, 102, 111, 112, 160 | N/A | N/A |
| dBET23 | 55, 57, 99, 102, 112 | 55, 57, 99, 102, 112 | N/A | N/A |
| dBET57 | 57, 55, 111, 112 | 55, 57, 72, 111, 112 | N/A | N/A |
| dBET70 | 55, 57, 99, 102, 112 | 55, 57, 99, 102 | N/A | N/A |

**Appendix 1—table 4.** PCA analysis with the first two PCs projection coverage in each degradation machinery complex.

TThe essential motions in the first two PCs of ten degradation machinery complexes were calculated based on the backbone atoms. The coverage of the first two PCs were reported. Note, all, expect A1_dBET1$_{\#35}$ and A1_dBET57$_{\#9}$ have the first two PCs coverage is below 60%.

| | A1 | | | B1 | | |
|---|---|---|---|---|---|---|
| PROTAC | PC1 | PC2 | PC1+PC2 | PC1 | PC2 | PC1+PC2 |
| dBET1$_{\#35}$ | 49.2% | 9.0% | 59.2% | 69.0% | 9.6% | 78.6% |
| dBET23$_{\#14}$ | 47.3% | 13.4% | 60.7% | 64.8% | 7.2% | 72.0% |
| dBET57$_{\#9}$ | 28.4% | 13.8% | 42.2% | 49.8% | 10.8% | 60.6% |
| dBET70$_{\#91}$ | 66.2% | 8.6% | 74.8% | 51.1% | 12.3% | 63.4% |
| dBET23$_{xray}$ | 50.7% | 11.9% | 62.6% | 50.7% | 15.4% | 66.1% |

