## [Editor Report · eLife Assessment]

This study provides **important** computational insights into the dynamics of PROTAC-induced degradation complexes, offering a **convincing** demonstration that differences in degradation efficacy can be linked to linker properties. The analyses address reproducibility considerations comprehensively, reinforcing the study's conclusions. Overall, these findings are significant for advancing cancer treatments and will be of broad interest to both biochemists and biophysicists.

---

## [Referee Report · Reviewer #1 (Public review)]

This study by Wu et al. provides valuable computational insights into PROTAC-related protein complexes, focusing on linker roles, protein-protein interaction stability, and lysine residue accessibility. The findings are significant for PROTAC development in cancer treatment, particularly breast and prostate cancers.

Strengths:

(1) Comprehensive computational analysis of PROTAC-related protein complexes.

(2) Focus on critical aspects: linker role, protein-protein interaction stability, and lysine accessibility.

Weaknesses:

(1) Limited examination of lysine accessibility despite its stated importance.

(2) Use of RMSD as the primary metric for conformational assessment, which may overlook important local structural changes.

The authors' claims about the role of PROTAC linkers and protein-protein interaction stability are generally supported by their computational data. However, the conclusions regarding lysine accessibility could be strengthened with more in-depth analysis. The use of the term "protein functional dynamics" is not fully justified by the presented work, which focuses primarily on structural dynamics rather than functional aspects.

Comments on revisions:

The authors have addressed the questions raised substantially.

---

## [Referee Report · Reviewer #2 (Public review)]

Summary:

The manuscript reports the computational study of the dynamics of PROTAC-induced degradation complexes. The research investigates how different linkers within PROTACs affect the formation and stability of ternary complexes between the target protein BRD4BD1 and Cereblon E3 ligase, and the degradation machinery. Using computational modeling, docking, and molecular dynamics simulations, the study demonstrates that although all PROTACs form ternary complexes, the linkers significantly influence the dynamics and efficacy of protein degradation. The findings highlight that the flexibility and positioning of Lys residues are crucial for successful ubiquitination. The results also discussed the correlated motions between the PROTAC linker and the complex.

Strengths:

The field of PROTAC discovery and design, characterized by its limited research, distinguishes itself from traditional binary ligand-protein interactions by forming a ternary complex involving two proteins. The current understanding of how the structure of PROTAC influences its degradation efficacy remains insufficient. This study investigated the atomic-level dynamics of the degradation complex, offering potentially valuable insights for future research into PROTAC degradability.

Comments on revisions:

All my questions have been addressed.

---

## [Referee Report · Reviewer #3 (Public review)]

The authors offer an interesting computational study on the dynamics of PROTAC-driven protein degradation. They employed a combination of protein-protein docking, structural alignment, atomistic MD simulations, and post-analysis to model a series of CRBN-dBET-BRD4 ternary complexes, as well as the entire degradation machinery complex. These degraders, with different linker properties, were all capable of forming stable ternary complexes but had been shown experimentally to exhibit different degradation capabilities. While in the initial models of the degradation machinery complex, no surface Lys residue(s) of BRD4 were exposed sufficiently for the crucial ubiquitination step, MD simulations illustrated protein functional dynamics of the entire complex and local side-chain arrangements to bring Lys residue(s) to the catalytic pocket of E2/Ub for reactions. Using these simulations, the authors were able to present a hypothesis as to how linker property affects degradation potency. They were able to roughly correlate the distance of Lys residues to the catalytic pocket of E2/Ub with observed DC50/5h values. This is an interesting and timely study that presents interesting tools that could be used to guide future PROTAC design or optimization.

---

## [Author Response]

The following is the authors’ response to the original reviews.

**Reviewer #1 (Public review):**
This study by Wu et al. provides valuable computational insights into PROTAC-related protein complexes, focusing on linker roles, protein-protein interaction stability, and lysine residue accessibility. The findings are significant for PROTAC development in cancer treatment, particularly breast and prostate cancers.The authors' claims about the role of PROTAC linkers and protein-protein interaction stability are generally supported by their computational data. However, the conclusions regarding lysine accessibility could be strengthened with more in-depth analysis. The use of the term "protein functional dynamics" is not fully justified by the presented work, which focuses primarily on structural dynamics rather than functional aspects.Strengths:(1) Comprehensive computational analysis of PROTAC-related protein complexes.(2) Focus on critical aspects: linker role, protein-protein interaction stability, and lysine accessibility.Weaknesses:(1) Limited examination of lysine accessibility despite its stated importance.(2) Use of RMSD as the primary metric for conformational assessment, which may overlook important local structural changes.
**Reviewer #1 (Recommendations for the authors):**
(1) The authors' claims about the role of PROTAC linkers and protein-protein interaction stability are generally supported by their computational data. However, the conclusions regarding lysine accessibility could be strengthened with more in-depth analysis. Expand the analysis of lysine accessibility, potentially correlating it with other structural features such as linker length.

We thank the reviewers for the suggestions! We performed time dependent correlation analysis to correlate the dihedral angles of the PROTACs and the Lys-Gly distance (Figures 6 and S17). We included detailed explanation on page 16:

“To further examine the correlation between PROTAC rotation and the Lys-Gly interaction, we performed a time-dependent correlation analysis. This analysis showed that PROTAC rotation translates motion over time, leading to the Lys-Gly interaction, with a correlation peak around 60-85 ns, marking the time of the interaction (Figure 6 and Figure S17). In addition, the pseudo dihedral angles also showed a high correlation (0.85 in the case of dBET1) with Lys-Gly distance. This indicated that degradation complex undergoes structural rearrangement and drives the Lys-Gly interaction.”

(2) The use of the term "protein functional dynamics" is not fully justified by the presented work, which focuses primarily on structural dynamics rather than functional aspects. Consider changing "protein functional dynamics" to "protein dynamics" to more accurately reflect the scope of the study.

Thanks to the reviewer for the suggestion to use the more accurate terminology! We agreed with the reviewer that if we keep “protein functional dynamics” in the title, we should focus on how the “overall protein dynamic” links to the “function” – The function is directly related to PROTAC-induced structural dynamics which is commonly seen in “protein-structural-function” relationship, but it is not our main focus. Therefore, we changed the title to replace “functional” by “structural”.

(3) Incorporate more local and specific characterization methods in addition to RMSD for a more comprehensive conformational assessment.

We thank the reviewer for the suggestion. We performed time dependent correlation analysis to understand how the rotation of PROTACs can translate to the Lys-Gly interaction. In addition, we performed dihedral entropies analysis for each dihedral angle in the linker of the PROTACs to better examine the flexibility of each PROTAC.

We included detailed explanation at page 18: “Our dihedral entropies analysis showed that dBET57 has ~0.3 kcal/mol lower entropies than the other three linkers, suggesting dBET57 is less flexible than other PROTACs (Figure S18).”

**Reviewer #2 (Public review):**
Summary:The manuscript reports the computational study of the dynamics of PROTAC-induced degradation complexes. The research investigates how different linkers within PROTACs affect the formation and stability of ternary complexes between the target protein BRD4BD1 and Cereblon E3 ligase, and the degradation machinery. Using computational modeling, docking, and molecular dynamics simulations, the study demonstrates that although all PROTACs form ternary complexes, the linkers significantly influence the dynamics and efficacy of protein degradation. The findings highlight that the flexibility and positioning of Lys residues are crucial for successful ubiquitination. The results also discussed the correlated motions between the PROTAC linker and the complex.Strengths:The field of PROTAC discovery and design, characterized by its limited research, distinguishes itself from traditional binary ligand-protein interactions by forming a ternary complex involving two proteins. The current understanding of how the structure of PROTAC influences its degradation efficacy remains insufficient. This study investigated the atomic-level dynamics of the degradation complex, offering potentially valuable insights for future research into PROTAC degradability.
**Reviewer #2 (Recommendations for the authors):**
(1) Regarding the modeling of the ternary complex, the BRD4 structure (3MXF) is from humans, whereas the CRBN structure in 4CI3 is derived from Gallus gallus. Is there a specific reason for not using structures from the same species, especially considering that human CRBN structures are available in the Protein Data Bank (e.g., 8OIZ, 4TZ4)?

We appreciate the reviewer’s insightful comment regarding the choice of crystal structures of BRD4 and CRBN structures from two species. Our initial selection of 4CI3 for CRBN structure was based on its high resolution and publication in Nature journal. Furthermore, the *Gallus gallus* CRBN structure shares high degree of sequence and structural similarity with *Homo sapiens* CRBN, especially in the ligand binding region. At the time of our study, we were aware of 4TZ4 as *Homo sapiens* CRBN, however, we did not use this structure since no publication or detailed experimental was associated with it. Additionally, PDB 8OIZ, was not publicly available yet for other researchers to use at the time.

(2) Based on the crystal structure (PDB ID: 6BNB) discussed in Reference 6, the ternary complex of dBET57 exhibits a conformation distinct from other PROTACs, with CRBN adopting an "open" conformation. Using the same CRBN structure for dBET57 as for other PROTACs might result in inaccurate docking outcomes.

Thank you for the reviewer’s comment! As noted by the authors in Reference 6, the observed open conformation of CRBN in the dBET57 ternary complex may result from the high salt crystallization conditions, which could drive structural rearrangement, and crystal contacts that may induce this conformation. The authors also mentioned that this open conformation could, in part, reflect CRBN’s intrinsic plasticity. However, they acknowledged that further studies are needed to determine whether this conformational flexibility is a characteristic feature of CRBN that enables it to accommodate a variety of substrates. Despite these observations, we believe that the compatibility of the observed BRD4^BD1^ binding conformation with both open and closed CRBN states suggests that these conformational changes are all possible. Therefore, we believe using the same initial CRBN structure for dBET57 as for other PROTACs can still reasonably reveal the dynamic nature of the ternary complex and would not significantly affect the accuracy of our docking outcomes either.

(3) Figure 2 displays only a single frame from the simulations, which might not provide a comprehensive representation. Could a contact frequency heatmap of PROTAC with the proteins be included to offer a more detailed view?

We thank the reviewer for the suggestion! We performed the contact map analysis to observe the average distance between PROTACs and BRD4^BD1^ over 400ns of MD simulation (new Figure S4 added).

We included detailed explanation at page 8 and 9: “The residues contact map throughout the 400ns MD simulation also showed different pattern of protein-protein interactions, indicating that the linkers were able to adopt different conformations (Figure S4).”

(4) The conclusions in Figure 3 and S11 are based on a single 400 ns trajectory. The reproducibility of these results is therefore uncertain.

We thank the reviewer for the suggestion! We added one more random seed MD simulation for each PROTAC to ensure the reproducibility of the results. The Result is shown in Figure S21 and the details for each MD run are updated in Table 1.

(5) Figure 4 indicates significant differences between the first and last 100 ns of the simulations. Does this suggest that the simulations have not converged? If so, how can the statistical analysis presented in this paper be considered reliable?

We thank the reviewers for the question. The simulation was initiated with a 10-15A gap between BRD4 and Ub to monitor the movement of degradation machinery and Lys-Gly interaction. The significant changes in pseudo dihedral in Figure 4 shows that the large-scale movement of the degradation complex can initiate the Lys-Gly binding. It does not relate to unstable sampling because the system remains very stable when BRD4 comes close to Ub.

(6) In Figure 5, the dihedral angle of dBET57_#9_MD1 is marked on a peptide bond. Shouldn't this angle have a high energy barrier for rotation?

We thank the reviewers for catching the error! Indeed, it was an error that the dihedral angles were marked on the peptide bond. We reworked the figure and double checked our dihedral correlation analysis. The updated correlate dihedral angle selection and the correlation coefficient is shown in Figure 5.

(7) Given that crystal structures for dBET 70, 23, and 57 are available, why is there a need to model the complex using protein-protein docking?

We thank the reviewer for the feedback. Only dBET23 has the ternary complex available in a crystal structure, which has the PROTAC and both proteins, while dBET1, dBET57 and dBET70 are not completed as ternary complexes. Although dBET70 has a crystal structure, its PROTAC’s conformation is not resolved, and thus we decided to still perform protein-protein docking with dBET70.

We includeed the explanation at page 8: “Only dBET23 crystal structure is available with the PROTAC and both proteins, while the experimentally determined ternary complexes of dBET1, dBET57 and dBET70 are not available. “

(8) On page 9, it is mentioned that "only one of the 12 PDB files had CRBN bound to DDB1 (PDB ID 4TZ4)." However, there are numerous structures of the DDB1-CRBN complex available, including those used for docking like 4CI3, as well as 4CI1, 4CI2, 8OIZ, etc.

We thank the reviewers for the comment! We acknowledged the existence of several DDB1-CRBN complex crystal structures, such as PDB IDs 4CI1, 4CI2, 4CI3, and the more recent 8OIZ. For our study, we chose to use 4TZ4 to maintain consistency in complex construction and to align with the methodology established in a previously published JBC paper (https://doi.org/10.1016/j.jbc.2022.101653), which successfully utilized the same structure for a similar construct. At the time our study was conducted, the 8OIZ structure had not yet been released. We appreciate your suggestion and will consider incorporating alternative structures in future studies to further investigate our findings.

(9) Table 2 is first referenced on page 8, while Table 1 is mentioned first on page 10. The numbering of these tables should be reversed to reflect their order of appearance in the text.

We thank the reviewer for catching the error! We switched the order of Table 1 and Table 2.

**Reviewer #3 (Public review):**
The authors offer an interesting computational study on the dynamics of PROTAC-driven protein degradation. They employed a combination of protein-protein docking, structural alignment, atomistic MD simulations, and post-analysis to model a series of CRBN-dBET-BRD4 ternary complexes, as well as the entire degradation machinery complex. These degraders, with different linker properties, were all capable of forming stable ternary complexes but had been shown experimentally to exhibit different degradation capabilities. While in the initial models of the degradation machinery complex, no surface Lys residue(s) of BRD4 were exposed sufficiently for the crucial ubiquitination step, MD simulations illustrated protein functional dynamics of the entire complex and local side-chain arrangements to bring Lys residue(s) to the catalytic pocket of E2/Ub for reactions. Using these simulations, the authors were able to present a hypothesis as to how linker property affects degradation potency. They were able to roughly correlate the distance of Lys residues to the catalytic pocket of E2/Ub with observed DC50/5h values. This is an interesting and timely study that presents interesting tools that could be used to guide future PROTAC design or optimization.
**Reviewer #3 (Recommendations for the authors):**
(1) My most important comment refers to the MM/PBSA analysis, the results of which are shown in Figure S9: binding affinities of -40 to -50 kcal/mol are unrealistic. This would correspond to a dissociation constant of 10^-37 M. This analysis needs to be removed or corrected.

We thank the reviewer for the comment! MM/PBSA analysis indeed cannot give realistic binding free energy. It does not include the configurational entropy loss which should be a large positive value. In addition, while the implicit PBSA solvent model computes solvation free energy, the absolute values may not be very accurate. However, because this is a commonly used energy calculation, and some readers may like to see quantitative values to ensure that the systems have stable intermolecular attractions, we kept the analysis in SI. We edited the figure legend, moved the Figure S10 in SI page 19, and added sentences to clearly state that the calculations did not include configuration entropy loss “Note that the energy calculations focus on non-bonded intermolecular interactions and solvation free energy calculations using MM/PBSA, where the configuration entropy loss during protein binding was not explicitly included. “.

(2) I think that the analysis of what in the different dBETx makes them cause different degradation potency is underdeveloped. The dihedral angle analysis (Figure 4B) did not explain the observed behavior in my opinion. Please add additional, clearer analysis as to what structural differences in the dBETx make them sample very different conformations.

We thank the reviewer for the suggestions! Based on the suggestion, we further performed dihedral entropy analysis for each dihedral angle in the linker part of the PROTAC to examine the flexibility of each PROTAC. Because each PROTAC has a different linker, we now clearly label them in a new Figure S18 in SI page 27. Low dihedral entropies indicate a more rigid structure and thus less flexibility to make a PROTAC more difficult to rearrange and facilitate the protein structural dynamic necessary for ubiquitination.

We added detailed explanation on page 18: “Our dihedral entropy analysis showed that dBET57 has ~0.3 kcal/mol lower configuration entropies than the other dBETs with three different linkers, suggesting that dBET57 is less flexible than the other PROTACs (Figure S18).”

(3) "The movement of the degradation machinery correlated with rotations of specific dihedrals of the linker region in dBETs (Figure 5).": this is not sufficiently clear from the figure. Definitely not in a quantitative way.

We thank the reviewers for the suggestions! To further understand the correlation between PROTACs dihedral angles and the movement of degradation machinery, we performed time dependent correlation analysis to correlate the dihedral angles of the PROTACs and the Lys-Gly distance (Figures 6 and S17).

We included detailed explanation on page 16:

“To further examine the correlation between PROTAC rotation and the Lys-Gly interaction, we performed a time-dependent correlation analysis. This analysis showed that PROTAC rotation translates motion over time, leading to the Lys-Gly interaction, with a correlation peak around 60-85 ns, marking the time of the interaction (Figure 6 and Figure S17). In addition, the pseudo dihedral angles also showed a high correlation (0.85 in the case of dBET1) with Lys-Gly distance. This indicated that degradation complex undergoes structural rearrangement and drives the Lys-Gly interaction.

(4) Cartoons are needed at multiple stages throughout the paper to enhance the clarity of what the modeled complexes looked like (e.g. which subunits they contained).

We thank the reviewers for the suggestions. We added and remade several Figures with cartoons to better represent the stages. We also used higher resolution and included clearer labels for each protein system.

(5) The difference between CRL4A E3 ligase and CRBN E3 ligase is not clear to the non-expert reader.

Thanks for the reviewer’s comment! To clarify the terms "CRL4A E3 ligase" and "CRBN E3 ligase", which refer to different levels of description for the protein complexes, we added a couple of sentences in the Figure 1 legend. As a result, the non-expert readers can clearly know the differences.

As illustrated in Figure 1,

CRL4A E3 ligase refers to the full E3 ligase complex, which includes all protein components such as CRBN, DDB1, CUL4A, and RBX1.CRBN E3 ligase, on the other hand, is a more colloquial term typically used to describe just the CRBN protein, often in isolation from the full CRL4A complex.

(6) Figure 1, legend: unclear why it's E3 in A and E2 in B.

We thank the reviewer for the question! E3 ligase in Figure 1A refers to CRBN E3 ligase, where researchers also simply term it CRBN. We have added a sentence to specify that CRBN E3 ligase is also termed CRBN for simplicity. In Figure 1B, E2 was unclear in the sentences. The full name of E2 should be E2 ubiquitin-conjugating enzyme. Because the name is a bit long, researchers also call it E2 enzyme. We have corrected it and used E2 enzyme to make it clearer.

(7) "Although the protein-protein binding affinities were similar, other degraders such as dBET1 and dBET57 had a DC50/5h of about 500 nM". It's unclear what experimental data supports the assertion that the protein-protein binding affinities are similar.

We thank reviewer for the question. Indeed, the statement is unclear.

We corrected the sentence in page 6: “Although utilizing the exact same warheads, other degraders such as dBET1 and dBET57 had a DC_50/5h_ of about 500 nM.”

(8) Was the construction of the degradation machinery complex guided by experimental data (maybe cryo-EM or tomography)? If not, what is the accuracy of the starting complex for MD? This may impact the reliability of the obtained results.

Thank you for your insightful comments! Yes, the construction of the degradation machinery complex was guided by available high-resolution crystal structures, which was selected to maintain consistency and align with the methodology established in a previously published JBC paper (https://doi.org/10.1016/j.jbc.2022.101653).

We acknowledged that static crystal structures represent only a single snapshot of the system and may not capture the full conformational flexibility of the complex. To address this limitation, we performed MD simulations using multiple starting structures. This approach allowed us to explore a broader conformational landscape and reduced the dependence on any single starting configuration, thereby enhancing the reliability of the results.

We hope this clarifies the robustness of our methodology and the steps taken to ensure accuracy in our simulations.

(9) "With quantitative data, we revealed the mechanism underlying dBETx-induced degradation machinery": I think this may be too strong of an assertion. The authors may have developed a mechanistic hypothesis that can be tested experimentally in the future.

We thank the reviewer for the suggestion. This is indeed a strong assertion and needs to be modified. We edited the sentence in page 7: “With quantitative data, we revealed the importance of the structural dynamics of dBETx-induced motions, which arrange positions of surface lysine residues of BRD4^BD1^ and the entire degradation machinery.”

(10) Figure S2: are the RMSDs calculated over all residues? Or just the BRD4 residues? Given that the structures are aligned with respect to CRBN, the reported RMSD numbers might be artificially low since there are many more CRBN residues than there are BRD4 residues. Also, why weren't the crystal structures used for dBET 23 and 70 for the modeling? Wouldn't you want to use the most accurate possible structures? Simulations were run for 23. Why not for 70?

We thank the reviewer for the suggestion. We added a sentence to more clearly explain the RMSD calculations in Figure S2: “The structural superposition is performed based on the backbone of CRBN and RMSD calculation is conducted based on the backbone of BRD4^BD1^.”

Although dBET70 has crystal structure, its PROTAC structure is not resolved, and thus we decided to still perform protein-protein docking with dBET70. dBET1 and dBET57 do not have a crystal structure for the ternary complexes.

We included the explanation at page 8: “Only dBET23 crystal structure is available with the PROTACs and both proteins, while the experimentally determined ternary complexes of dBET1, PROTACs of dBET57 and dBET70 are not available. “

a. And there are no crystal structures available for 1 and 57? If so, please clearly say that. Otherwise please report the RMSD.

We thank the reviewer for the suggestion. We included the explanation at page 8: “Only dBET23 crystal structure is available with the PROTACs and both proteins, while the experimentally determined ternary complexes of dBET1, PROTACs of dBET57 and dBET70 are not available.”

(11) Table 2 is referenced before Table 1.

We thank the reviewer for catching the error! We switched the order for Table 1 and Table 2.

(12) Figure S3 is not referenced in the main paper.

We thank the reviewer for catching the error! We now referred Figure S3 on page7.

(13) Minor comments on grammar and sentence structure:a. It should be "binding of a ternary complex"b. "Our shows the importance": word missing.c. "...providing insights into potential orientations for ubiquitination. observe whether the preferred conformations are pre-organized for ubiquitination." Word or words missing.

We thank reviewer for catching the errors! We corrected grammatical errors and unclear sentences throughout the entire paper and revised the sentences to make them easily understandable for non-expert readers.